# Targeting translation initiation yields fast-killing therapeutics against the zoonotic parasite *Cryptosporidium parvum*

Meng Li[1©], Jigang Yin[1©], Dongqiang Wang[1], Beibei Zou[1], Guan Zhu[1,2]*

**1** State Key Laboratory for Diagnosis and Treatment of Severe Zoonotic Infectious Diseases, Key Laboratory for Zoonosis Research of the Ministry of Education, Institute of Zoonosis, and College of Veterinary Medicine, Jilin University, Changchun, Jilin, China, **2** Department of Veterinary Pathobiology, Texas A and M University, College Station, Texas, United States of America

© These authors contribute equally to the work.
\* cryptosporida@gmail.com

## Abstract

*Cryptosporidium parvum* is a zoonotic apicomplexan that causes moderate-to-severe watery diarrhea in children, immunocompromised patients, and neonatal ruminants, yet no fully effective drug is available. We show that the parasite's eukaryotic initiation factor 4A (CpeIF4A; a DEAD-box RNA helicase in the eIF4F translation-initiation complex) can be exploited as a fast-killing therapeutic target. Rocaglamide A (Roc-A), a plant-derived rocaglate, binds the CpeIF4A–RNA–ATP complex with high affinity ($K_d = 33.7$ nM) and blocks protein synthesis in excysting sporozoites ($IC_{50} \approx 3.7$ μM). In host-cell culture, Roc-A suppresses intracellular growth with nanomolar potency ($EC_{50} = 1.77$ nM) and a selectivity index exceeding 56,000 in HCT-8 cells and 1,400 in HepG2 cells. A five-day oral regimen (0.5 mg/kg/d) reduced oocyst shedding by >90% within 48 h in interferon-γ–knockout mice and by 70–90% from day 2 onward without rebound during a 15-day follow-up in NCG mice. Two amino-acid differences at the Roc-A binding surface (D165 and V192 in CpeIF4A vs. N167 and D194 in the human ortholog) offer a foothold for medicinal optimization toward greater parasite selectivity. These findings establish translation initiation as an unexplored but tractable pathway for anti-cryptosporidial drug discovery and position Roc-A as a promising lead compound.

## Author summary

*Cryptosporidium parvum* is a common intestinal parasite that can cause life-threatening diarrhea in young children, people with weakened immune systems, and newborn calves. Current treatments are only partly effective, so new drugs are urgently needed. Protein translation, the process that turns mRNA into protein, has not been explored as a drug target in this parasite. Here we

**Data availability statement:** All relevant data are within the manuscript and its Supporting Information files.

**Funding:** This work was funded by a grant from National key research and development program (award number 2023YFD1801000 to G.Z.). The funders had no role in study design, data collection and analysis, decision to publish, or preparation of the manuscript.

**Competing interests:** The authors have declared that no competing interests exist.

identify CpeIF4A, an RNA-unwinding enzyme that kick-starts translation, as a weak point. We show that the natural product rocaglamide A (Roc-A) binds tightly to CpeIF4A when the enzyme is clamped onto mRNA, shutting down protein synthesis and killing the parasite quickly. Roc-A works at low-nanomolar concentrations in cell culture and, when given orally to two different immunodeficient mouse models, cuts oocyst shedding by 70–90% within two days and keeps parasite levels low for more than two weeks after treatment stops. Because key residues in the rocaglamide binding pocket differ between the parasite and humans, the compound can be modified to enhance selectivity and reduce toxicity. Our results open the door to a new class of fast-acting drugs against cryptosporidiosis.

## Introduction

*Cryptosporidium* is a genus of enteric apicomplexan parasites that are widely distributed worldwide. Of more than 40 named species, *C. parvum* and *C. hominis* cause the vast majority of human cryptosporidiosis cases [1]. In immunocompetent individuals the infection is usually acute and self-limiting, with diarrhea that lasts about one to two weeks. In vulnerable populations (e.g., infants, young children, and people with compromised immunity), the disease often becomes prolonged, severe, and sometimes fatal [2,3]. In resource-limited regions, cryptosporidiosis ranks among the leading causes of pediatric diarrhea and is linked to growth stunting and increased mortality [3–5]. The zoonotic species *C. parvum* also infects neonatal ruminants and some other animals, especially calves, causing major economic losses and acting as a reservoir for human infection [6–9]. Therapeutic options remain limited, as nitazoxanide is the only drug approved by the U.S. Food and Drug Administration, and its efficacy is incomplete in immunocompromised patients [10,11]. Consequently, new anti-cryptosporidial agents are urgently needed for both medical and veterinary use [11,12].

Drug discovery for *Cryptosporidium* faces several technical and biological obstacles [10]. The parasite does not complete its life cycle under routine in vitro conditions, and although genetic tools now exist, they remain cumbersome for broad application [13,14]. The organism's streamlined metabolism deletes many classical targets: it lacks an apicoplast, a canonical mitochondrial Krebs cycle and electron-transport chain, and de novo biosynthetic pathways for nucleosides, amino acids, or fatty acids [15,16]. Additionally, *Cryptosporidium* resides on top of enterocytes, enclosed by a host-derived parasitophorous vacuole membrane (PVM) yet separated from the host cytoplasm by an electron-dense band and a membrane structure called the feeder organelle (FO) [17,18]. These interface structures are selectively permeable and can hinder the access of certain small molecules to the parasite, adding another layer of complexity to drug delivery.

In target-based drug discovery, the past decade has seen escalating efforts to identify selective inhibitors of enzymes and factors that drive the core metabolic

pathways still retained by *Cryptosporidium* (e.g., energy metabolism, fatty-acid activation and elongation, nucleotide synthesis, protein synthesis, and protein modification) [19–23]. These campaigns have yielded a number of anti-cryptosporidial hits and leads, including some compounds with well-defined inhibitory kinetics, nanomolar in vitro potency, high selectivity indices, and encouraging in vivo efficacy [11,24]. Representative examples include triacsin C, an inhibitor of *C. parvum* fatty-acyl-CoA synthetase (ACS) [25]; vorinostat, which targets histone deacetylase (HDAC) [26]; bumped-kinase inhibitors such as BKI-1708 and BKI-1553 that block calcium-dependent protein kinase 1 (CDPK1) [27]; KDU731, a phosphatidylinositol-4-kinase (PI4K) inhibitor [23]; and "compound 5," which inhibits lysyl-tRNA synthetase (KRS) [22]. All of these targets participate in fundamental metabolic or regulatory processes that are essential for parasite growth and development.

Here we demonstrate that the translation-initiation machinery of *Cryptosporidium*, which was previously unexplored in anti-cryptosporidial drug discovery, can be pharmacologically exploited. We focus on *C. parvum* eukaryotic initiation factor 4A (CpeIF4A), a DEAD-box RNA helicase that forms part of the eIF4F complex (eIF4A–eIF4E–eIF4G). Rocaglamide A (Roc-A), a selective plant-derived rocaglate from *Aglaia* species, binds and inhibits the function of CpeIF4A (Fig 1A) [28]. Inhibiting CpeIF4A blocks protein synthesis in excysting sporozoites, suppresses intracellular parasite growth at low-nanomolar concentrations in vitro, and sharply reduces oocyst shedding in interferon-γ–knockout and NCG mice. Together, these results establish CpeIF4A as a novel anti-cryptosporidial target and position Roc-A as a promising lead for potential therapeutic development and structure-guided optimization.

## Results

### The eIF4A inhibitor Roc-A is highly efficacious and selective against the in-vitro growth of *C. parvum* and acts directly on the parasite target

To explore whether translation initiation machinery could serve as a viable drug target in *C. parvum*, we first tested the in vitro anti-cryptosporidial efficacy of Roc-A, a selective inhibitor of eIF4A critical to the translational initiation [28]. Using our standard 44-h in vitro assay, in which *C. parvum* was cultured in HCT-8 cells with parasite loads evaluated by qRT-PCR at 44 h post-infection (hpi) [29,30], Roc-A showed single-digit nanomolar potency (i.e., the 50% effective concentration [$EC_{50}$] = 1.77 nM) (Fig 1B). In contrast, cytotoxicity of Roc-A on the host cells was minimal, as HCT-8 cells were viable up to 100 μM ($TC_{50} > 100$ μM) (Fig 1C), giving a selectivity index (SI) of >56,000. The HepG2 cells were somewhat more sensitive ($TC_{50} = 2.54$ μM), yet still provided an SI of ~1,400.

The excellent selectivity suggested that Roc-A acts directly on the parasite, rather than via a host cell pathway [31], which was confirmed using our recently developed MDR1-overexpression model. In this model, HCT-8 cells were trans-fected to overexpress the multi-drug resistance-1 gene (*MDR1*) (HCT-8/MDR1) in comparison to the blank vector negative control (HCT-8/NC). If a compound is an MDR1 substrate, MDR1 overexpression would increase drug tolerance to the compound. When the compound acts fully on the epicellular *Cryptosporidium*, the increased drug tolerance in host cells would not alter the anti-cryptosporidial efficacy [31,32]. Here, transient overexpression of MDR1 was used (close to 100% transfection efficiency as shown in S1 Fig), which showed no effect on the growth of HCT-8 cells (Fig 1D, bars 1a vs. 1b), but increased the tolerance to Roc-A (Fig 1D, bars 2a vs. 2b). Elacridar, a selective MDR1 inhibitor, reversed this tolerance in a dose-dependent manner (Fig 1D, bars from 3a to 6b), verifying that Roc-A is an MDR1 substrate. Impor-tantly, MDR1 overexpression had no effect on the anti-cryptosporidial activity of Roc-A (i.e., $EC_{50} = 2.35$ nM or 2.30 nM in HCT-8/MDR1 or HCT-8/NC cells, respectively; p = 0.9586 by multiple comparison test) (Fig 1E), excluding a host-target mechanism.

Next, we examined activity against free sporozoites and early intracellular stages. Viability assay based on qRT-PCR detection of 18S rRNA showed that excysted sporozoites survive up to 2 h in PBS at 25°C (Fig 2A). Within this window, Roc-A inhibited the sporozoite viability with an $EC_{50}$ of 54 nM (Fig 2B), which is ~30-fold higher than in the 44-h infection

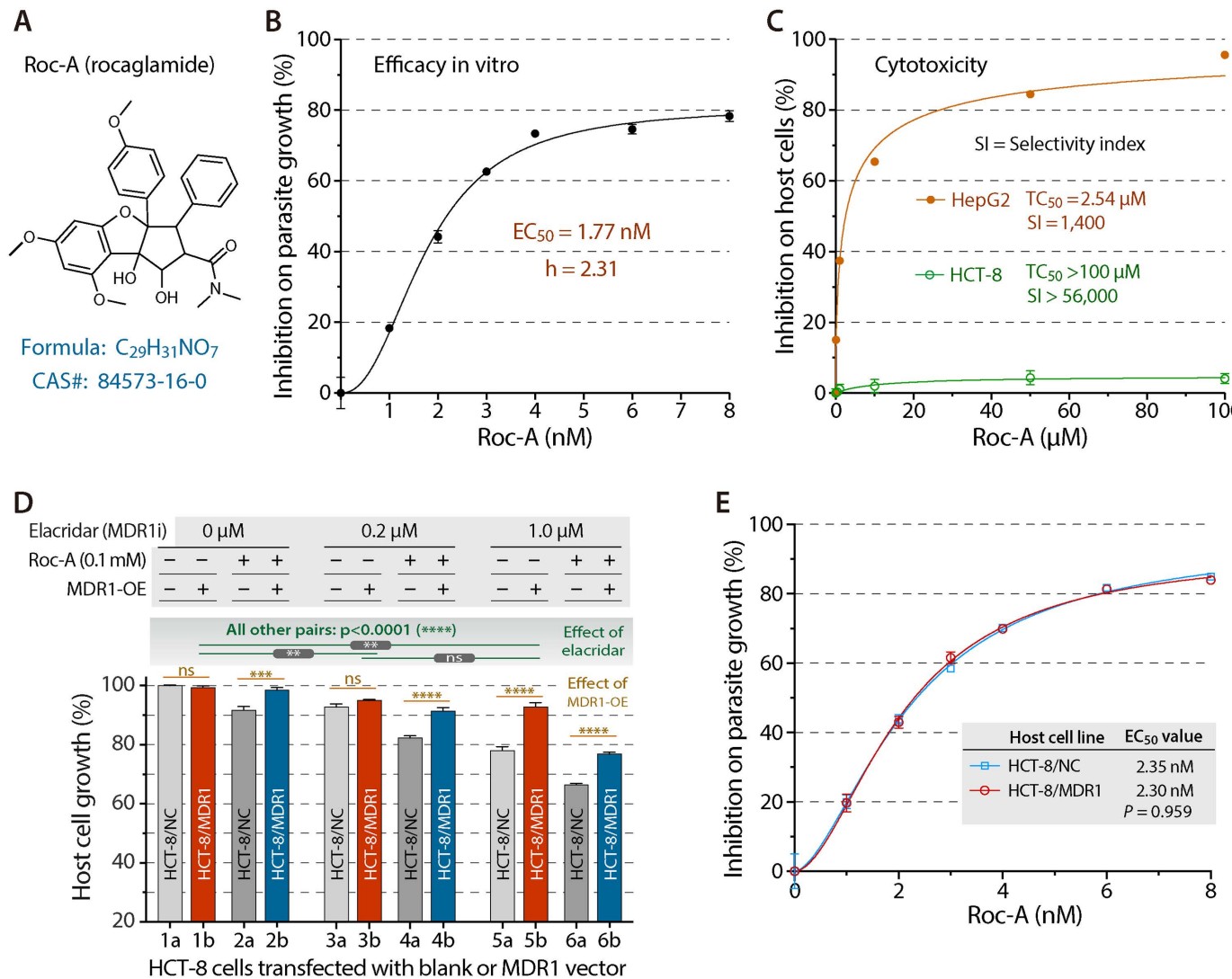

**Fig 1. Rocaglamide A (Roc-A) selectively inhibits *C. parvum* growth by acting on the parasite eIF4A target. (A)** Chemical structure of Roc-A (CAS 84573-16-0), a highly selective inhibitor of eukaryotic initiation factor 4A (eIF4A). **(B)** Dose-response curve for Roc-A in a 44-h infection assay: *C. parvum* was grown in HCT-8 monolayers and parasite burden quantified by qRT-PCR targeting 18S rRNA (Cp18S). $EC_{50} = 1.77$ nM. **(C)** Host-cell cytotoxicity. HCT-8 and HepG2 cells were exposed to Roc-A for 44 h and viability measured by MTS. Selectivity index (SI) = $TC_{50}/EC_{50}$. **(D)** Confirmation that Roc-A is an MDR1 substrate. HCT-8 cells transiently over-expressing human MDR1 (HCT-8/MDR1) or vector control (HCT-8/NC) were treated with 0.1 µM Roc-A in the absence or presence of the MDR1 inhibitor elacridar. MDR1 over-expression increased host-cell tolerance to Roc-A in every elacridar group. Significance by Tukey's multiple comparisons: ** $p < 0.01$, *** $p < 0.001$, **** $p < 0.0001$; ns, not significant. **(E)** Over-expression of MDR1 does not alter Roc-A efficacy against *C. parvum*: inhibitory curves and $EC_{50}$ values are identical in HCT-8/MDR1 and HCT-8/NC monolayers, demonstrating that the antiparasitic effect is on-target within the parasite.

assay. By comparison, paromomycin displayed an $EC_{50}$ of 280.7 µM, which is > 5,000-fold less potent. In the "early-stage" assay (Fig 2C), cultures were treated with Roc-A at 3 nM (~$EC_{65}$ in the 44-h infection assay) during invasion (0–3 hpi), early development (3–10 hpi), or the full 44 h (3–44 hpi). A 3-nM treatment during invasion slightly reduced parasite load by 6.7% (p < 0.01; vs. 0.5% DMSO control), whereas a 7-h post-invasion treatment cut load by 52.6% (p < 0.0001). Continuous exposure reduced burden by 70.5%, consistent with the $EC_{50}$ curve (62.7% at 3 nM; Fig 1B).

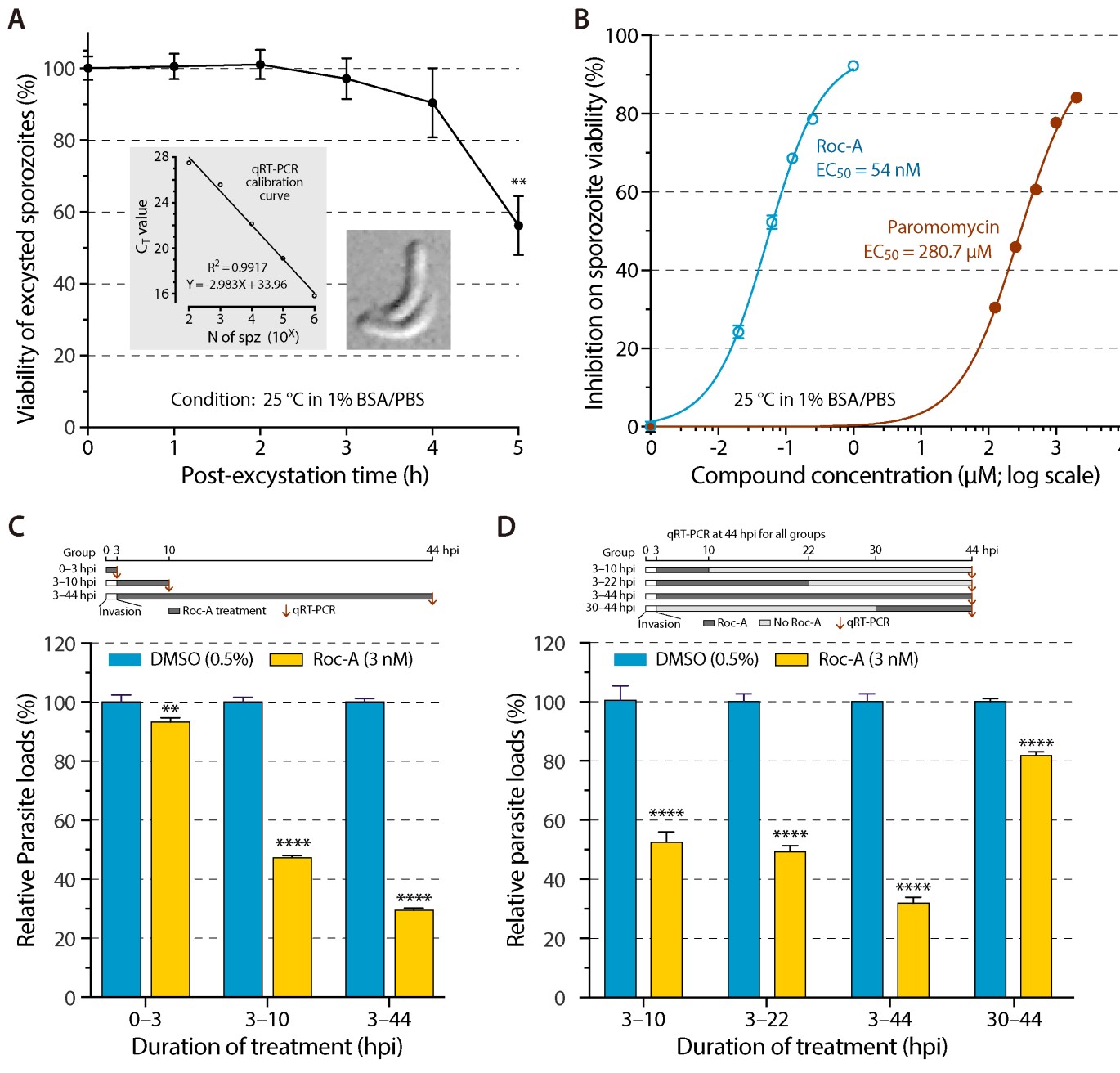

**Fig 2. Rocaglamide A (Roc-A) reduces sporozoite viability and blocks invasion and early intracellular development of *C. parvum*. (A)** Time-dependent viability of free sporozoites held at 25°C in PBS (pH 7.4) containing 1% BSA. Viability was measured by qRT-PCR for *C. parvum* 18S rRNA (Cp18S). Insets: calibration curve ($C_T$ vs. sporozoite number) and representative phase-contrast image of two excysted sporozoites. **(B)** Viability of excysted sporozoites after 2 h exposure to Roc-A or paromomycin (PMM) at the indicated concentrations (1% BSA/PBS, 25°C), quantified by Cp18S qRT-PCR. **(C)** Effect of Roc-A (3 nM in 0.5% DMSO; ~$EC_{63}$) during invasion (0–3 hpi) or first-generation merogony (3–10 hpi) compared with continuous treatment (3–44 hpi). Parasite burden was measured by qRT-PCR at the end of each exposure; schematic timeline is shown above the graph. **(D)** Drug-withdrawal assay distinguishing parasiticidal from parasitistatic action. Following invasion, cultures received Roc-A (3 nM) for 3–10 hpi or 3–22 hpi; drug was then removed and parasites were allowed to develop until 44 hpi. A continuous 3–44 hpi treatment served as reference; timeline is illustrated above the graph. For panels **C** and **D**, oocysts were added to HCT-8 monolayers, allowed to excyst and invade for 3 h, and uninvaded parasites were washed away before the indicated treatments. Statistical significance was determined by Sidak's multiple comparisons test (** $p < 0.01$, **** $p < 0.0001$ versus the 0.5% DMSO control at the same time point).

Drug withdrawal experiment clarified whether Roc-A is parasiticidal (Fig 2D). Treatment with 3 nM Roc-A for 7 h (3–10 hpi) or 19 h (3–22 hpi) yielded 47.4% and 50.5% reductions, respectively, with no rebound after drug removal. The full-course treatment (3–44 hpi) reduced the parasite loads by 68.1%, which is comparable to the 70.5% and 62.7% reductions in the two experiments described above (Figs 1B and 2C). In contrast, exposure limited to the sexual stage (30–44 hpi) lowered burden by only ~20%. The reduced activity late in development may reflect lower protein-synthesis rates, as both macrogametes and microgametes are relatively inactive and unable to fertilize to form zygotes under in vitro condition, and/or reduced drug permeability. Together, the data indicate that Roc-A rapidly and irreversibly kills asexual stages of *C. parvum* in vitro.

In summary, Roc-A is an exceptionally potent and selective inhibitor of *C. parvum* growth in vitro. Its antiparasitic effect is fully attributable to engagement of the parasite target, and the compound acts as a fast-killing, parasiticidal agent.

## Roc-A is highly efficacious in vivo and acts rapidly against chronic *C. parvum* infection in IFN-γ-KO and NCG mice

We first determined a tolerated dose in uninfected C57BL/6 mice. A single daily oral dose of Roc-A at 0.5 mg/kg for 7 days caused no adverse effect on body-weight gain or health scores (S2 Fig). The same 0.5 mg/kg dose, administered as two split gavages of 0.25 mg/kg (12 h apart), was therefore used for all subsequent studies.

The first efficacy experiment employed IFN-γ–KO mice on a C57BL/6 background and ran for 15 days (n = 15; mixed gender). Mice were infected with one dose of *C. parvum* ($5 \times 10^4$ oocysts/ mouse) and allowed to establish chronic infection for 10 days (Fig 3A). Beginning at 10 day post-infection (dpi), designated 0 day post-administration (dpa), animals received Roc-A, paromomycin (PRM, 1,000 mg/kg/d), or vehicle. In vehicle-treated controls, oocyst shedding peaked on 10 dpi and then declined gradually, reaching a 48.9% reduction by 5 dpa (Table 1; Fig 3B). PRM produced a slower, moderate decline of 36.1–71.0% over the same interval. Roc-A, by contrast, induced a rapid 90.9–97.5% reduction within 24 h of the first dose and maintained that suppression through 5 dpa. Body-weight gains did not differ significantly among the three groups, although Roc-A-treated mice showed a transient dip on 2 dpa followed by faster recovery than either control (Fig 3C).

A second experiment used severely immunodeficient NCG mice and extended the observation period to 30 days (n = 15; mixed gender). Infection and treatment protocols matched those of the IFN-γ-KO study, with oocyst shedding and body-weight gains monitored for 15 days after the 5-day treatment (Fig 3A). Vehicle controls again peaked at 10 dpi, dipped during treatment, and exhibited a secondary peak at 17 dpi before gradually decreasing (Table 2; Fig 3D). Roc-A reduced shedding by 73.3% on 1 dpa and by 90.4–94.4% on days 2–5. Importantly, after drug withdrawal oocyst shedding remained 92.2–95.8% below baseline throughout the 15-day follow-up, indicating a parasiticidal rather than parasitistatic effect. Roc-A also improved body-weight gain and composite health scores relative to vehicle (both p < 0.0001 by two-way ANOVA; Fig 3E, 3F).

Taken together, the two mouse studies demonstrate that Roc-A is markedly more effective than PRM, achieves >90% suppression of oocyst shedding within a day of treatment onset, and maintains this effect without rebound after dosing stops. These in vivo data corroborate the fast-killing, parasiticidal activity observed in vitro and further validate translation initiation, specifically CpeIF4A, as a promising therapeutic target for cryptosporidiosis.

## Roc-A targets the cytoplasmic translation-initiation factor 4A of *C. parvum* (CpeIF4A) with low-nanomolar activity

The *C. parvum* genome encodes an eIF4A (gene ID cgd1_880; GenBank XP_001388148) that contains the sequence hallmarks of the eIF4A subfamily, including an N-terminal DEAD-box helicase domain (InterPro IPR044728), a C-terminal helicase domain (IPR001650), and conserved residues for ATP binding as well as interaction with the eIF4G subunit (Fig 4A, 4B). In addition, the genome harbors a closely related gene annotated as eIF4A1 (cgd7_3940; XP_628555) together

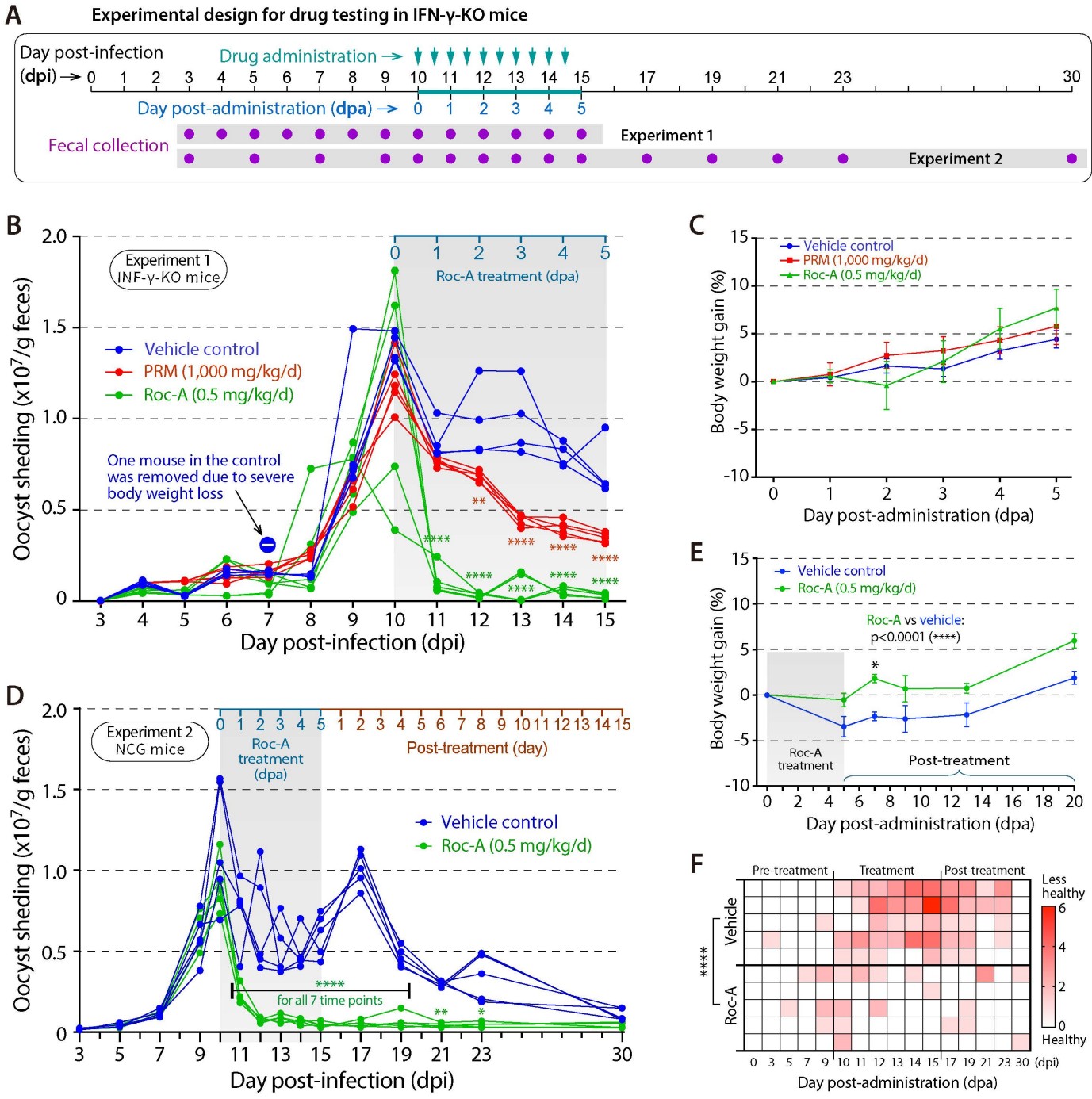

**Fig 3. Rocaglamide A (Roc-A) is highly efficacious in two murine models of chronic *C. parvum* infection. (A)** Schematic of the dosing protocol: IFN-γ-knockout (IFN-γ-KO) or NCG mice were infected orally with $5 \times 10^4$ oocysts, allowed to establish infection for 10 days, and then treated twice daily for 5 days with Roc-A (0.5 mg/kg/d), paromomycin (PRM, 1,000 mg/kg/d; IFN-γ-KO study only), or vehicle (1% DMSO). **(B, C)** IFN-γ-KO study (n = 5 per group). Roc-A reduced oocyst shedding by >90% within 24 h of the first dose and sustained suppression through day 5 **(B)**, while body-weight trends were comparable to vehicle and PRM groups **(C)**. One vehicle-treated mouse was removed on day 3 owing to excessive weight loss. **(D–F)** NCG study (n = 5 per group). Roc-A again produced a rapid, ≥ 90% decline in oocyst shedding that persisted without rebound through the 15-day follow-up **(D)**; treated mice gained more weight **(E)** and had higher composite health scores **(F)** than vehicle controls. Oocyst output was quantified by qPCR, calibrated against standard curves generated by spiking known oocyst numbers into negative feces and processing identically. Statistical differences versus vehicle were assessed by Tukey's multiple comparisons test (** p < 0.01; **** p < 0.0001). Error bars denote SEM.

**Table 1. Relative levels of fecal oocyst shedding in C57BL/7 interferon-γ knockout mice chronically infected with *C. parvum*.**

| Baseline for normalization | DPA* | DPI | Vehicle (1% DMSO) | | | PRM (1000 mg/kg/d) | | | Roc-A (0.5 mg/kg/d) | | |
|---|---|---|---|---|---|---|---|---|---|---|---|
| | | | Mean | SEM | N | Mean | SEM | N | Mean | SEM | N |
| In each group (vs. 0 dpa) | 0 | 10 | 100.0 | 2.9 | 4 | 100.0 | 5.5 | 5 | 100.0 | 22.6 | 5 |
| | 1 | 11 | 62.9 | 3.7 | 4 | 63.9 | 0.9 | 5 | 9.1 | 2.8 | 5 |
| | 2 | 12 | 70.1 | 7.3 | 4 | 57.0 | 1.1 | 5 | 2.7 | 0.5 | 5 |
| | 3 | 13 | 71.2 | 7.1 | 4 | 37.1 | 1.2 | 5 | 5.4 | 3.0 | 5 |
| | 4 | 14 | 57.5 | 2.4 | 4 | 33.6 | 1.5 | 5 | 5.0 | 0.9 | 5 |
| | 5 | 15 | 51.1 | 5.7 | 4 | 29.0 | 1.0 | 5 | 2.5 | 0.5 | 5 |
| On each day (vs. vehicle) | 0 | 10 | 100.0 | 2.9 | 4 | 85.9 | 4.8 | 5 | 86.4 | 19.5 | 5 |
| | 1 | 11 | 100.0 | 6.0 | 4 | 87.4 | 1.2 | 5 | 12.6 | 3.9 | 5 |
| | 2 | 12 | 100.0 | 10.5 | 4 | 69.9 | 1.3 | 5 | 3.3 | 0.6 | 5 |
| | 3 | 13 | 100.0 | 10.0 | 4 | 44.7 | 1.4 | 5 | 6.5 | 3.7 | 5 |
| | 4 | 14 | 100.0 | 4.1 | 4 | 50.3 | 2.2 | 5 | 7.5 | 1.4 | 5 |
| | 5 | 15 | 100.0 | 11.2 | 4 | 48.8 | 1.7 | 5 | 4.2 | 0.9 | 5 |

* Roc-A (0.5 mg/kg/d) and vehicle were given in split doses (0.25 mg/kg gavages 12 h apart) for five days from 10 to 15 DPI. **Abbreviations:** DPA, day post-administration of compounds; DPI, day post-infection. DMSO, dimethyl sulfoxide; PRM, paromomycin; Roc-A, rocaglamide-A.

**Table 2. Relative levels of fecal oocyst shedding in NCG mice chronically infected with *C. parvum*.**

| Baseline for normalization | DPA* | DPI | Vehicle control (1% DMSO) | | | Roc-A (0.5 mg/kg/d) | | |
|---|---|---|---|---|---|---|---|---|
| | | | Mean | SEM | N | Mean | SEM | N |
| In each group (vs. 0 dpa) | 0 | 10 | 100.0 | 14.4 | 5 | 100.0 | 9.2 | 5 |
| | 1 | 11 | 64.8 | 6.1 | 5 | 26.7 | 2.7 | 5 |
| | 2 | 12 | 54.2 | 7.3 | 5 | 9.6 | 1.0 | 5 |
| | 3 | 13 | 46.8 | 6.9 | 5 | 8.2 | 0.8 | 5 |
| | 4 | 14 | 42.9 | 3.2 | 5 | 7.4 | 0.8 | 5 |
| | 5 | 15 | 55.2 | 5.4 | 5 | 5.6 | 0.6 | 5 |
| | 7 | 17 | 85.7 | 6.0 | 5 | 5.6 | 0.9 | 5 |
| | 9 | 19 | 37.7 | 1.8 | 5 | 7.8 | 2.0 | 5 |
| | 11 | 21 | 28.2 | 1.2 | 5 | 4.9 | 0.4 | 5 |
| | 13 | 23 | 24.6 | 1.1 | 5 | 5.5 | 0.9 | 5 |
| | 20 | 30 | 10.9 | 1.4 | 5 | 4.2 | 0.5 | 5 |
| On each day (vs. vehicle control) | 0 | 10 | 100.0 | 14.4 | 5 | 100.0 | 9.2 | 5 |
| | 1 | 11 | 100.0 | 9.4 | 5 | 41.1 | 4.1 | 5 |
| | 2 | 12 | 100.0 | 13.5 | 5 | 17.8 | 1.8 | 5 |
| | 3 | 13 | 100.0 | 14.7 | 5 | 17.6 | 1.6 | 5 |
| | 4 | 14 | 100.0 | 7.5 | 5 | 17.3 | 1.8 | 5 |
| | 5 | 15 | 100.0 | 9.7 | 5 | 10.1 | 1.0 | 5 |
| | 7 | 17 | 100.0 | 7.0 | 5 | 6.5 | 1.1 | 5 |
| | 9 | 19 | 100.0 | 4.7 | 5 | 20.8 | 5.3 | 5 |
| | 11 | 21 | 100.0 | 4.1 | 5 | 17.4 | 1.5 | 5 |
| | 13 | 23 | 100.0 | 4.4 | 5 | 22.2 | 3.6 | 5 |
| | 20 | 30 | 100.0 | 13.3 | 5 | 38.9 | 4.3 | 5 |

* Roc-A (0.5 mg/kg/d) and vehicle were given in split doses (0.25 mg/kg gavages 12 h apart) for five days from 10 to 15 DPI. **Abbreviations:** DPA, day post-administration of compounds; DPI, day post-infection. DMSO, dimethyl sulfoxide; Roc-A, rocaglamide-A.

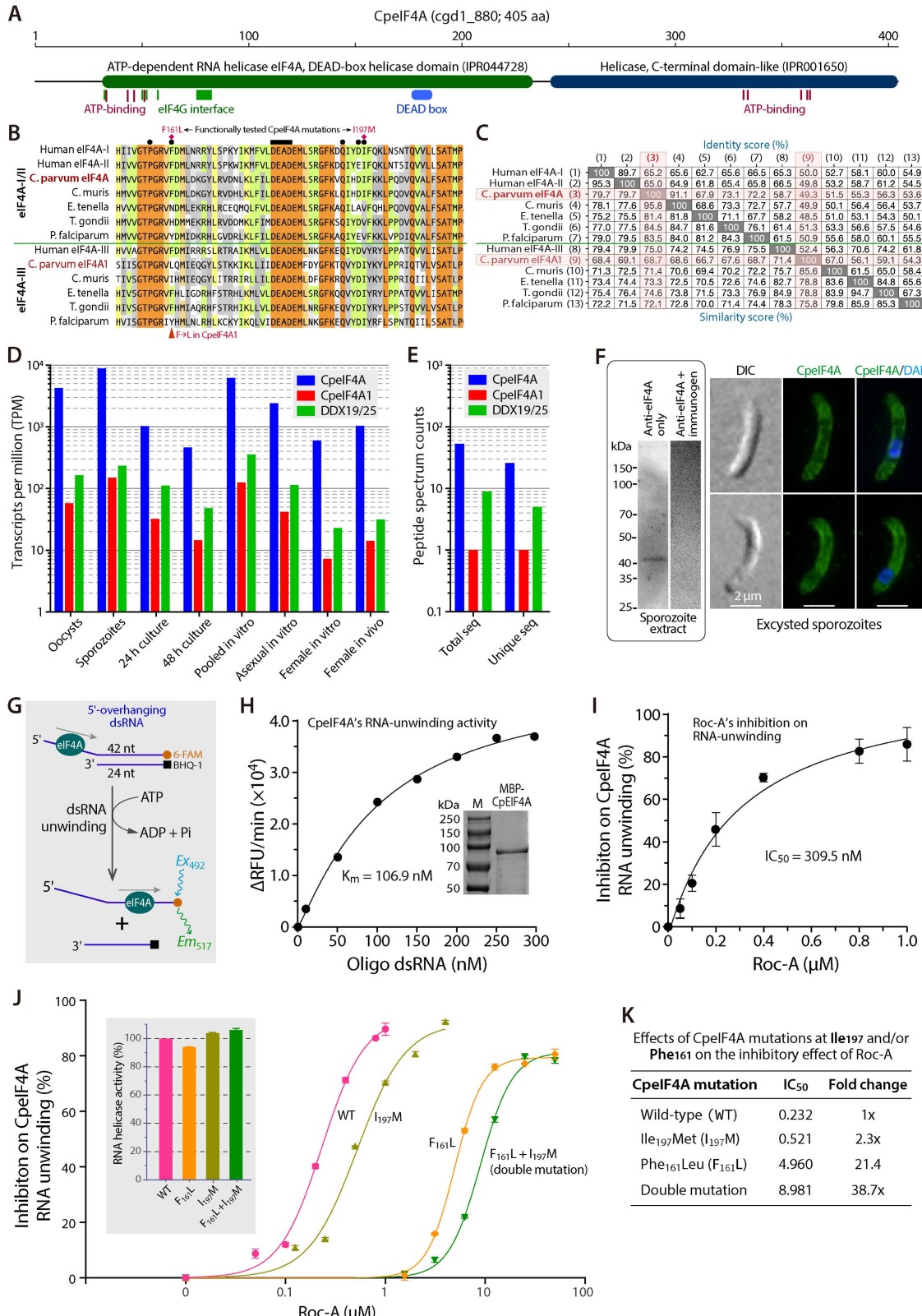

**Fig 4. Molecular and biochemical characterization of *C. parvum* eIF4A (CpeIF4A). (A)** Domain architecture of CpeIF4A (cgd1_880; 405 aa) predicted by InterProScan, showing N-terminal and C-terminal helicase domains, ATP-binding motifs, and eIF4G-contact residues. **(B)** Alignment of the

conserved DEAD-box region from CpeIF4A, human eIF4A-I/II, orthologs from *C. muris*, *E. tenella*, *T. gondii*, and *P. falciparum*, and the eIF4A-III–type proteins (CpeIF4A1, human eIF4A-III, and their apicomplexan orthologs). The bar marks the canonical DEAD motif; filled circles denote Roc-A contact residues (PDB 5ZC9). Red diamonds indicate positions mutated in this study. **(C)** Pairwise identity and similarity scores for the sequences in panel **B. (D, E)** Relative abundance of CpeIF4A versus CpeIF4A1 across parasite stages, extracted from CryptoDB transcriptomes (D) and proteomes **(E)**; CpeIF4A is expressed ≥25-fold higher at every stage (see S1–S2 Tables). **(F)** Detection of native CpeIF4A. Left, western blot of sporozoite lysate probed with an affinity-purified rabbit anti-CpeIF4A peptide antibody; a single band appears at the expected size. Right, immunofluorescence shows cytoplasmic localization with stronger signal beneath the plasma membrane (green); nuclei are counter-stained with DAPI (blue). DIC, differential-interference-contrast image. **(G–I)** RNA-unwinding assay for recombinant MBP-CpeIF4A (scheme in **G**). Michaelis–Menten plot for duplex RNA (H; inset: SDS-PAGE of purified protein) and dose–response curve for Roc-A inhibition (I). **(J)** Effect of point mutations Phe161Leu (F161L), Ile197Met (I197M), and the double mutant on helicase activity (inset) and binding affinity to Roc-A. **(K)** Corresponding shifts in Roc-A sensitivity: $IC_{50}$ increases 2.3-fold (I197M), 21.4-fold (F161L), and 38.7-fold (double mutant), indicating Phe161 is the predominant determinant of high-affinity binding.

with 23 other DEAD-box (DDX) RNA helicases, giving 25 DDX proteins in total. Nineteen of these retain the fully conserved "DEAD" motif, whereas five display DExD and one DEAx variants. Among them, CpeIF4A shares the highest identity with the human eIF4A-I/II isoforms that drive cytoplasmic translation initiation [28,33]; CpeIF4A1 aligns most closely with human eIF4A-III (Fig 4C), which operates primarily in the nucleus as part of the exon-junction complex [34,35]. Bayesian-inference phylogeny, built from DDX sequences of *Cryptosporidium* and other apicomplexans, placed CpeIF4A with human eIF4A-I/II and grouped CpeIF4A1 with eIF4A-III (S3 Fig), confirming these assignments.

Transcriptomic and proteomic data in CryptoDB show that the CpeIF4A gene is expressed at far higher levels than CpeIF4A1 throughout the asexual cycle: transcript abundance is 31.9- to 82.1-fold higher and peptide abundance 26- to 115-fold higher, depending on the sample (Fig 4D, 4E; S1 and S2 Tables). A rabbit polyclonal antibody raised against a CpeIF4A-specific peptide detected a single band of the expected size in sporozoite lysate by western blotting (Fig 4F, left), confirming specificity. Immunofluorescence revealed a cytoplasmic distribution with slightly stronger signal beneath the plasma membrane in both free sporozoites (Fig 4F, right) and intracellular stages (S4 Fig). Taken together, the expression profile, localization, and phylogeny support CpeIF4A as the housekeeping translation-initiation helicase in the parasite.

Recombinant CpeIF4A expressed as a maltose-binding-protein (MBP) fusion exhibited low-nanomolar RNA-unwinding activity in a fluorescence assay ($K_m = 106.9$ nM) (Fig 4G, 4H). Roc-A inhibited this activity with an $IC_{50}$ of 309.5 nM (Fig 4I). Thermal-shift analysis showed that Roc-A bound CpeIF4A with an apparent dissociation constant ($K_d$) of 3.51 µM (Fig 5A); the affinity increased more than 100-fold to 33.7 nM when 5'-overhanging double-stranded RNA (dsRNA) and ATP were present (Fig 5B). By comparison, dsRNA and ATP alone bound CpeIF4A with $K_d$ values of 1.13 µM and 625.7 µM, respectively (Fig 5C, 5D). Thus, Roc-A interacts far more tightly with the CpeIF4A–RNA–ATP complex than with the apo enzyme, consistent with the structural model for human eIF4A-I in which Roc-A clamps the helicase onto polypurine sequences of mRNA and blocks initiation [36]. The relatively modest affinity for free CpeIF4A likely facilitates subsequent formation of the high-affinity ternary complex.

### Structural modeling and mutagenesis support high-affinity binding of Roc-A to the CpeIF4A–RNA complex and point to avenues for improving specificity

To clarify how Roc-A interacts with CpeIF4A in the presence of RNA, we built a homology model using the crystal structure of human eIF4A bound to RNA and Roc-A (PDB 5CZ9) as the template and SWISS-MODEL for refinement (Figs 6A and S5). The resulting structure superimposes on the human template with an RMSD of 0.08 Å (Fig 6B), indicating very high overall similarity. At the Roc-A interface, the two residues critical for ligand binding in the human enzyme, Phe163 and Ile199, are conserved in CpeIF4A, as are D164, Q196, D198, and K202 (Fig 6B, black labels).

Three positions differ: N167, R168, and D194 in human eIF4A correspond to D165, K166, and V192 in CpeIF4A (Fig 6B, red labels). Two of these changes alter both identity and physicochemical character; acidic D165 replaces neutral N167, and non-polar V192 replaces acidic D194. Because D165 and V192 lie within 5 Å of the ligand, modifications that enhance hydrophobic or electrostatic contacts with these residues could increase Roc-A selectivity for the parasite enzyme.

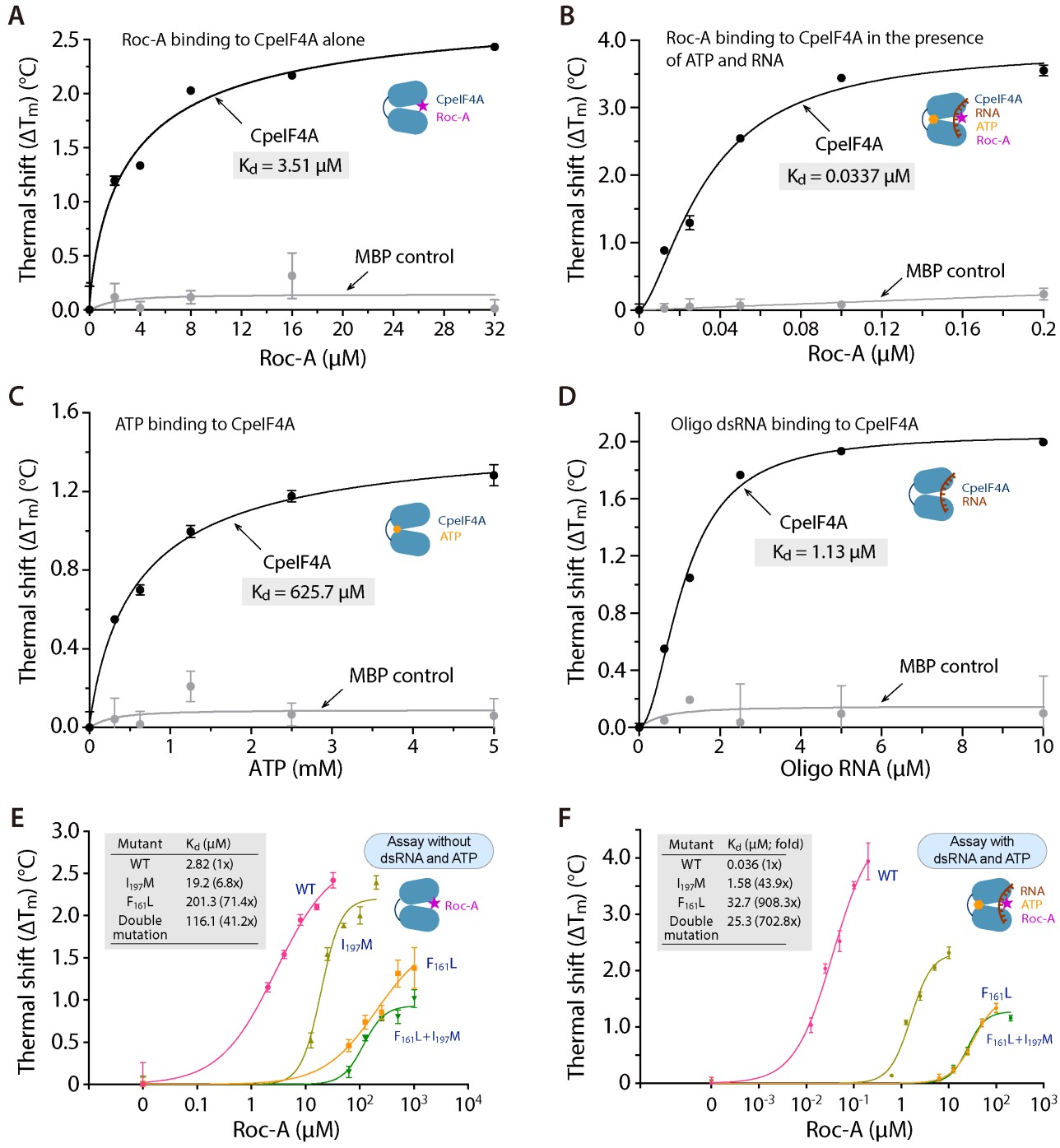

**Fig 5. Rocaglamide A (Roc-A) binds CpeIF4A with highest affinity when duplex RNA and ATP are present. (A–B)** Thermal-shift assay (TSA) of Roc-A binding to apo-CpeIF4A (A) and to CpeIF4A pre-incubated with duplex RNA and ATP **(B)**. The substrates tighten the interaction, lowering the apparent $K_d$ from 3.51 µM to 33.7 nM ($\approx 10^2$-fold increase in affinity). **(C, D)** Control TSA curves for CpeIF4A binding to the individual substrates ATP (C) and duplex RNA **(D)**. **(E, F)** Effect of point mutations F161L, I197M, and the double mutant (F161L+I197M) on Roc-A affinity in the absence (E) or presence (F) of RNA+ATP. I197M raises the $K_d$ ~7-fold, F161L ~71-fold, and the double mutant ~41-fold in the apo state. When RNA+ATP are included, the losses deepen to ~44-, ~908-, and ~703-fold, respectively, confirming that Phe161 is the principal determinant of high-affinity binding.

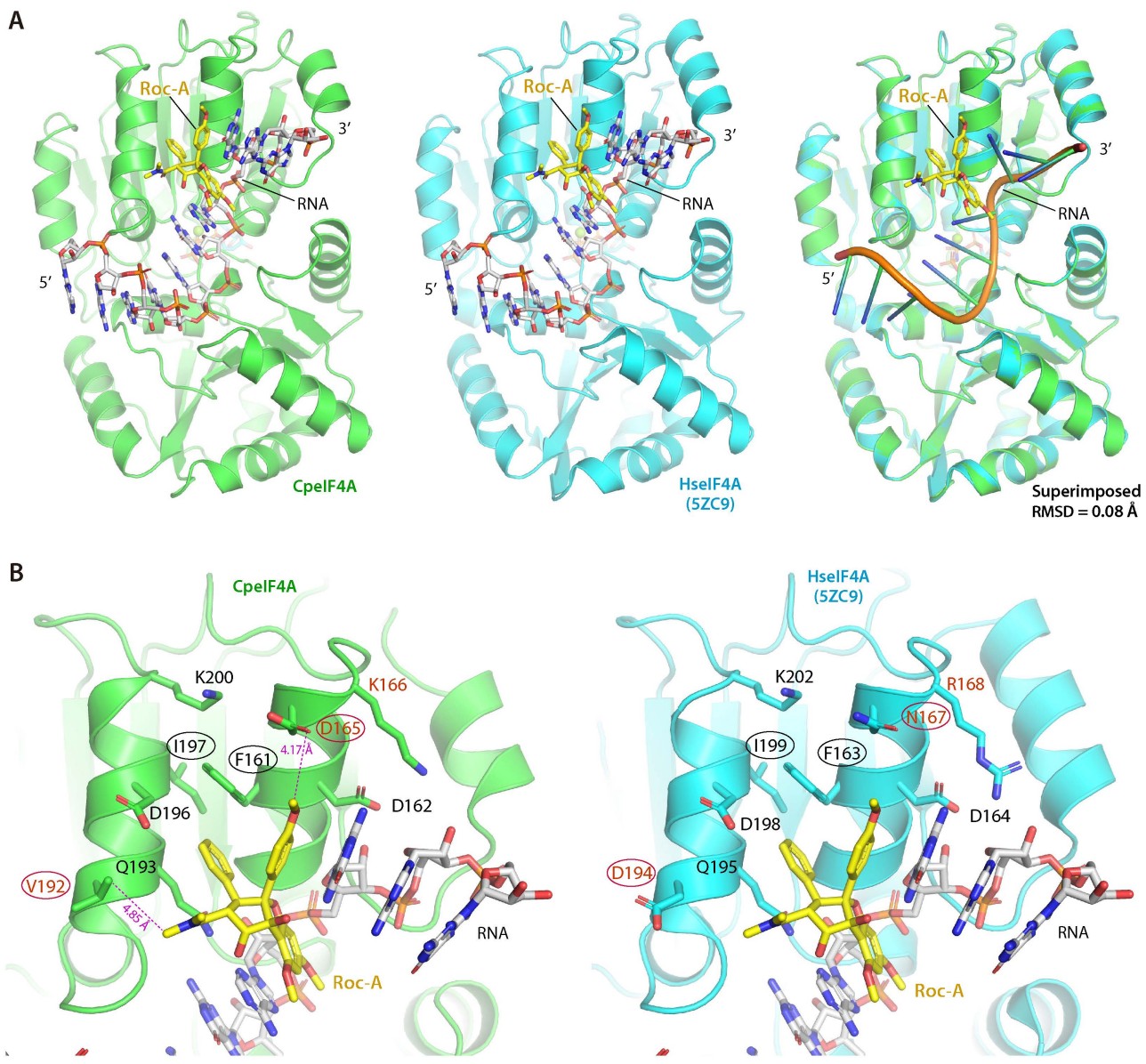

**Fig 6. Structural comparison of Roc-A binding to *C. parvum* eIF4A (CpeIF4A) and human eIF4A (HseIF4A). (A)** Homology model of CpeIF4A (left) built on the crystal structure of the HseIF4A–AMP-PNP–Roc-A–polypurine RNA complex (center; PDB 5ZC9). Right, superposition of the two structures illustrates their overall similarity. **(B)** Close-up of the Roc-A–binding pocket in CpeIF4A (left) and HseIF4A (right). Residues essential for ligand contact are labeled; conserved positions are in black, parasite-specific substitutions in red. Black-circled residues (Phe161, Ile197) were validated by mutagenesis, whereas the red-circled substitutions (Asp165, Val192) represent sites that could be exploited to enhance Roc-A selectivity for the parasite enzyme.

The importance of the binding surface was tested by substituting Phe161 and Ile197, which correspond to Leu and Met in the rocaglate-producing tree *Aglaia odorata* [36]. Single mutants F161L and I197M and the double mutant F161L/I197M showed only minor changes in basal RNA-helicase activity (Fig 4J, inset) but large losses in Roc-A sensitivity. $IC_{50}$ values rose 2.3-fold for I197M, 21.4-fold for F161L, and 38.7-fold for the double mutant (Fig 4J). Binding affinities measured by thermal-shift assay shifted accordingly: in the absence of ATP and RNA, $K_d$ increased 6.8-fold, 71.4-fold, and 41.2-fold for I197M, F161L, and F161L/I197M, respectively (Fig 5E). With ATP and RNA present, the corresponding increases were

43.9-fold, 908.3-fold, and 702.8-fold (Fig 5F). These data confirm that both Phe161 and Ile197 contribute to high-affinity Roc-A binding, with Ile197 playing the dominant role.

### Inhibition of CpeIF4A by Roc-A suppresses protein synthesis in excysting sporozoites while leaving global transcription largely unchanged

Blocking translation initiation should repress protein synthesis, so we measured nascent protein production in sporozoites that emerge during oocyst excystation, a stage that can be obtained in pure form. Pilot tests showed that Roc-A, at concentrations up to 100 µM, did not interfere with excystation or with sporozoite viability (Fig 7A). In these assays oocysts were incubated in excystation medium for 2 h, a window that permits additional protein synthesis yet preserves viability in the absence of host cells. Under the same conditions, Roc-A (≤ 10 µM) did not alter the sub-cellular distribution of CpeIF4A (Fig 7B). Because sporozoites remain protected by the oocyst wall until the suture opens, and then exit rapidly (Fig 7C), Roc-A is unlikely to affect the excystation process itself; instead, the two-hour window is suitable for assessing its effect on translation.

Protein synthesis was quantified by western blotting with a pan-*Cryptosporidium* antiserum raised against total oocyst proteins. Roc-A reduced band intensities in sporozoite pellets (non-secreted proteins) in a concentration-dependent manner, yielding an $IC_{50}$ of 3.65 µM (Fig 7D, left panel; Fig 7E, blue line). Secreted proteins in the supernatant were largely unaffected, except for a prominent 28-kDa band that declined with a similar $IC_{50}$ of 3.80 µM (Fig 7D, right panel; Fig 7E, red line). The matched inhibitory kinetics indicate that de-novo synthesis of both the bulk cytoplasmic proteins and this 28-kDa secreted protein is CpeIF4A-dependent and sensitive to Roc-A. Because secreted proteins are mostly pre-synthesized and stored in sporozoites before release, the stronger effect on non-secreted proteins is expected. Overall, these data support a model in which Roc-A, by binding CpeIF4A, blocks the parasite's translation-initiation machinery and suppresses protein synthesis.

In contrast, Roc-A produced minimal transcriptional changes. Free sporozoites were treated with 150 nM Roc-A (≈ $EC_{70}$; Fig 2B) or 0.5% DMSO for 2 h at 25°C, and RNA-seq was performed on three biological replicates. Transcripts from 3,795 protein-coding genes were detected, and 3,783 genes had reads in both conditions. Using q-value (adjusted P-value) of 0.05 and a two-fold cutoff, no differentially expressed genes (DEGs) were observed (Fig 7F). Relaxing the threshold to an unadjusted P = 0.05 and $|\log_2 FC| > 2$ identified only seven DEGs (0.185%): four down-regulated and three up-regulated genes of unknown function, all expressed at low levels (relative abundances 7.5–32.4 in controls and 0.7–56.2 in Roc-A–treated samples, compared with median values of 617–625 and mean values of ~4,500 transcripts; Fig 7F; S3 Table). Thus, Roc-A profoundly inhibits protein synthesis without broadly altering the sporozoite transcriptome, consistent with its primary action on the translation-initiation factor CpeIF4A.

## Discussion

This study shows that the translational machinery of *C. parvum* can be disrupted by targeting eukaryotic initiation factor 4A (eIF4A). Roc-A is highly potent in vitro ($EC_{50}$ = 1.77 nM) and, in vivo, a five-day course at 0.5 mg/kg/d lowered oocyst shedding by more than 90% within 48 h in IFN-γ-KO mice and by 70–90% from day 2 onward, with no rebound during a 15-day follow-up, in NCG mice. Using an HCT-8 model that over-expresses MDR1, we confirmed that parasite killing is an on-target effect. Roc-A acts rapidly and irreversibly; it inhibits the helicase activity of the cytoplasmic CpeIF4A ($IC_{50}$ = 309.5 nM) through tight binding to the CpeIF4A–RNA–ATP complex ($K_d$ = 33.7 nM) and suppresses protein synthesis in excysting sporozoites, chiefly for non-secreted proteins ($IC_{50}$ ≈ 3.7 µM).

Roc-A also binds human DDX3 with lower affinity [37]. It may likewise engage additional DEAD-box (DDX) helicases in *C. parvum*, creating synergistic effects. Sequence comparison points to CpeIF4A1 as a plausible secondary target: this protein resembles human eIF4A-III, is predicted to reside in both cytoplasm and nucleus, and carries a Phe → Leu substitution at the Roc-A binding site that markedly lowers drug affinity in mutagenesis tests (Figs

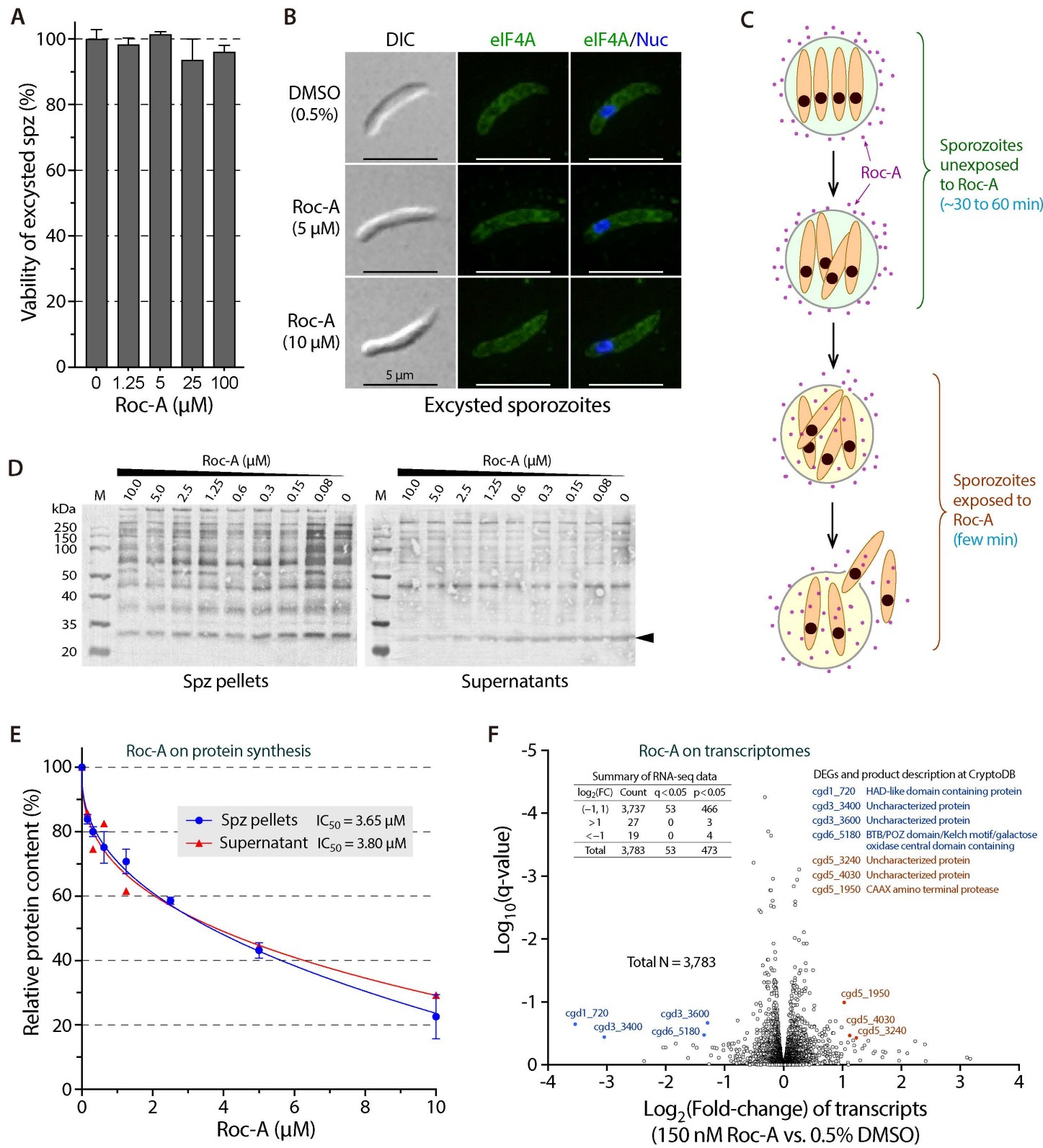

**Fig 7. Rocaglamide A (Roc-A) suppresses protein synthesis but not global transcription in excysting *C. parvum* sporozoites.** **(A)** Viability of sporozoites released in vitro for 2 h in the presence of 0–100 µM Roc-A. Cp18S qRT-PCR shows no loss of viability at any concentration tested. **(B)**

Morphology and CpeIF4A localization after 2 h exposure to 5 or 10 µM Roc-A. Sporozoites were stained with affinity-purified anti-CpeIF4A (green) and DAPI (blue); no changes are evident. **(C)** Schematic of excystation. Roc-A can reach sporozoites only after the oocyst suture ruptures and parasites exit. **(D)** Western blot of non-secreted (pellet) and secreted (supernatant) proteins from sporozoites excysted for 2 h with increasing Roc-A. A pan-*Cryptosporidium* antibody detects dose-dependent loss of most pellet bands, whereas only a single 28-kDa secreted band (arrowhead) declines. **(E)** Quantification of band intensity versus Roc-A concentration. Blue, total non-secreted proteins; red, 28-kDa secreted protein. Error bars, SEM. **(F)** Volcano plot of RNA-seq data from sporozoites treated for 2 h with 0.15 µM Roc-A (~$EC_{70}$) versus 0.5% DMSO. Of 3,783 transcripts detected, none meet the stringent threshold |$\log_2 FC$| > 2, $q < 0.05$; only seven low-abundance, uncharacterized genes change under the relaxed criterion |$\log_2 FC$| > 2, $p < 0.05$.

4J, and. 5F). Because CpeIF4A1 is expressed at much lower levels than CpeIF4A (Fig 4D, 4E), its contribution to parasite killing is probably minor. The parasite genome also encodes two prohibitins (cgd7_4240 and cgd5_2320), homologous to mammalian and *Plasmodium* PHB1/2, which have been reported as rocaglate targets [38,39]. Although translation appears to be the dominant pathway affected, the roles of these additional proteins merit investigation.

eIF4A is a linchpin for protein synthesis in several protozoa. In *P. falciparum*, eIF4A co-localizes with eIF4E throughout asexual blood stages, and RNA interference of either subunit halts growth [40]. Cap-pulldown proteomics confirmed eIF4A as a core component of the *Plasmodium* initiation apparatus and suggested stage-specific regulation [41]. Trypanosoma-tids possess distinct eIF4F-like complexes. In *Trypanosoma brucei*, the single eIF4A ortholog binds one complex under normal conditions, and a catalytic mutant shifts binding and loses translational activity [42]. Similar rewiring occurs in *T. cruzi*, where nutritional stress redirects eIF4A-containing complexes to specific mRNA pools [43]. In *Leishmania* spp., structure-guided screens produced ATP-site inhibitors that discriminate between parasite and human eIF4A, validating the target [44].

Rocaglates such as Roc-A, silvestrol, and CR-1-31B inhibit several protozoa. A cross-kingdom survey identified low-nanomolar sensitivity in *P. falciparum*, *T. gondi*i, and *Trypanosoma* spp., whereas species lacking the eIF4A motif were refractory [45]. CR-1-31B cures experimental malaria at 0.2–0.5 mg/kg and also dampens host inflammation [46]. Roc-A engages *P. falciparum* PHB1/2, blocking both asexual growth and transmission, including artemisinin-resistant strains [39].

Susceptibility is not universal. *Trypanosoma brucei* and *Leishmania infantum* are 10–100 times less sensitive than mammalian cells because a phenylalanine-to-valine change weakens binding [42]. An *Ophiocordyceps* fungus that colonizes rocaglate-producing *Aglaia* trees carries a Gly→Phe reversal that abolishes drug binding [47], contrasting with our finding that Phe161 is more critical than Ile197 for CpeIF4A binding.

Despite variable cross-species activity, the low-nanomolar $EC_{50}$ and high selectivity index (>1,400) indicate that *C. parvum* is far more sensitive than host cells. In mice, however, the safety margin narrows; toxicity appears at 1.5 mg/kg/day, while 0.5 mg/kg/day is well tolerated. These data validate CpeIF4A as a promising drug target and suggest that Roc-A is a useful lead. Two parasite-specific substitutions at the Roc-A interface (N167→D165 and D194→V192; vs. mammalian eIF4A) create opportunities for medicinal chemistry to enhance selectivity. Roc-A also serves as a probe to dissect the poorly understood regulation of protein synthesis in *Cryptosporidium*.

An unexpected finding is that Roc-A strongly suppresses protein synthesis in excysting sporozoites but leaves the transcriptome essentially unchanged (Fig 7F). One explanation is a delay between translation arrest and detectable effects on mRNA abundance; pre-existing transcription factors may sustain basal transcription until they are depleted. This delay would be most evident during excystation, when sporozoites remain within intact oocysts for 30–60 min before rapid release [48]. Alternatively, newly excysted sporozoites may require little additional mRNA, synthesizing primarily proteins needed for immediate invasion. These possibilities are biologically plausible but require experimental confirmation.

In summary, our data establish CpeIF4A as a druggable Achilles' heel in *C. parvum* and underscore the potential of rocaglates as fast-acting, parasiticidal agents against cryptosporidiosis.

## Materials and methods

### Ethics statement

All animal procedures complied with the Guide for the Care and Use of Laboratory Animals of the Ministry of Health (China) and were approved by the Animal Welfare and Research Ethics Committee of Jilin University (AUP No. 2020-1Z-20).

### Parasite and host cells

An isolate of *C. parvum* (*gp60* subtype IIaA17G2R1) was propagated in calves and purified from feces by sequential sucrose and CsCl gradient centrifugation, following published protocols [49,50]. Purified oocysts were stored at 4°C in PBS supplemented with 200 U/mL penicillin and 0.2 mg/mL streptomycin. Viability was checked routinely by excystation; only batches with > 80% excystation were used. Before host-cell infection, oocysts were treated with 10% bleach (v/v) on ice for 5 min, washed five times with PBS, and resuspended in culture medium.

HCT-8 cells were maintained in RPMI 1640, and HepG2 cells in DMEM; both media contained 10% fetal bovine serum (FBS), 50 U/mL penicillin, and 50 µg/mL streptomycin. Cultures were incubated at 37°C in a humidified atmosphere with 5% $CO_2$.

### In vitro drug-efficacy assay

Roc-A activity against *C. parvum* was measured with a "44-h infection assay" [29,30]. The protocol comprises 3 h of excystation/invasion followed by 41 h of intracellular development in the continuous presence of test compound. HCT-8 monolayers (~80% confluence) were prepared in 96-well plates. Bleach-treated oocysts were suspended in RPMI-1640 containing 0.15% taurocholic acid and added by medium exchange ($5 \times 10^4$ oocysts/well). After 3 h, uninvaded parasites were removed by exchanging with RPMI-1640 plus 10% FBS and antibiotics; Roc-A was added at the indicated concentrations (0.5% DMSO diluent control). Plates were incubated for 41 h at 37°C, 5% $CO_2$.

At 44 h post-infection (hpi), plates were centrifuged and supernatants discarded. Ice-chilled 0.1% bovine serum albumin (BSA, nuclease-free) was added (50 µL/well) and plates were vortexed on ice for 30 min [29,30]. Lysates were cleared by centrifugation (2,000 g, 15 min) and used immediately or stored at –80°C. Parasite burden was quantified by one-step SYBR Green qRT-PCR (HiScript II, Vazyme) on a StepOnePlus system (Applied Biosystems). Each 20 µL reaction contained 2 µL lysate (1:100 dilution), 0.4 µL of each primer (10 µM), 1 µL enzyme mix, 0.4 µL ROX reference dye, 10 µL 2×SYBR mix, and nuclease-free water. Primers were Cp18S-1011F (5′-TTGTTCCTTACTCCTTCAGCAC-3′) and Cp18S-1185R (5′-TCCTTCCTATGTCTGGACCTG-3′) for *C. parvum* 18S rRNA, and Hs18S-1F (5′-GGCGCCCCCTCGA TGCTCTTA-3′) and Hs18S-1R (5′-CCCCCGGCCGTCCTCTTA-3′) for human 18S rRNA.

Growth inhibition was calculated by the $\Delta\Delta C_T$ method, and $EC_{50}$ values were obtained by fitting a four-parameter sigmoidal curve [29,30]. Standard curves were generated by infecting HCT-8 cells with graded oocyst numbers and processing identically.

### In vitro cytotoxicity assay

The cytotoxic effect of Roc-A was determined with an MTS cell-viability assay (Saint-Bio, Shanghai, China). HCT-8 and HepG2 cells ($1 \times 10^4$ cells/well) were seeded in 96-well plates and allowed to attach for 16 h. Serial dilutions of Roc-A (final DMSO, 0.5% v/v) were then added, and the cultures were incubated for 41 h. Wells were rinsed three times with PBS, after which 10 µL of MTS reagent was added to each well and the plates were incubated at 37°C for 2 h. Absorbance at 490 nm was read on a Synergy LX multimode plate reader (BioTek, Winooski, VT). Percent viability was calculated as

$$Viability\ (\%) = \frac{OD_{490(\text{treat})} - OD_{490(\text{blank})}}{OD_{490(\text{control})} - OD_{490(\text{blank})}} \times 100$$

where the control received 0.5% DMSO. $TC_{50}$ values for HCT-8 and HepG2 cells were obtained from dose–response curves; each concentration was tested in triplicate or greater. The in-vitro selectivity index (SI) was calculated as $TC_{50}/EC_{50}$.

### Drug effects on excysted sporozoites, early intracellular stages, and drug-withdrawal assay

The impact of Roc-A on *C. parvum* sporozoites, invasion, and early intracellular development was assessed by qRT-PCR detection of parasite 18S rRNA. Oocysts were excysted in PBS (pH 7.4) containing 0.75% taurocholic acid at 37°C for 45 min, yielding >90% excystation. Excysted sporozoites were pelleted and resuspended in PBS or RPMI-1640. Because sporozoites survive only briefly without host cells, we first quantified their viability over time. Aliquots of $8 \times 10^5$ sporozoites in PBS plus 1 mg/mL BSA were incubated at 25°C for 0–5 h (three replicates per time point); pellets were lysed in 0.1% BSA–water (50 µL) and Cp18S rRNA was measured by qRT-PCR. Viability remained >80% for at least 2 h (Fig 2A), so a 2-h window was adopted for compound testing.

For the "sporozoite assay," sporozoites in 1% BSA/PBS were exposed for 2 h at 25°C to serial dilutions of Roc-A or paromomycin (0.5% DMSO final). Sporozoites were then pelleted, lysed, and Cp18S rRNA quantified. $EC_{50}$ values were derived from standard curves generated with known sporozoite numbers.

Early intracellular effects were examined in HCT-8 monolayers (~80% confluence). Roc-A at 3 nM (approximately the $EC_{65}$ from the 44-h assay; Fig 1B) was applied during invasion (0–3 hpi) or first-generation merogony (3–10 hpi). A full 3–44 hpi exposure served as reference (Fig 2C). Parasite burden was determined by qRT-PCR at the end of each treatment.

To distinguish parasitistatic from parasiticidal action, a drug-withdrawal protocol was used. After invasion, cultures received Roc-A (3 nM) either from 3–10 hpi or 3–22 hpi; drug was then removed by medium exchange and parasites were allowed to develop until 44 hpi. A continuous 3–44 hpi group provided comparison, and a 30–44 hpi groups was included to assess the effect of Roc-A on sexual development (Fig 2D). Parasite loads for all groups were quantified at 44 hpi by qRT-PCR. Each condition in all in-vitro assays was tested with three or more biological replicates.

### Validation of on-target activity with MDR1-overexpressing host cells

To verify that Roc-A acts directly on the parasite rather than through a host process, we used an MDR1-overexpression model that tests whether a compound's efficacy drops when host cells become more drug-tolerant [31,32]. HCT-8 cells were transiently transfected with pCMV3-MDR1 (HCT-8/MDR1); cells receiving the empty vector served as controls (HCT-8/NC). Immunofluorescence with an anti-MDR1 antibody (ab170904, Abcam, Cambridge, UK) confirmed ~100% transfection efficiency under identical staining and exposure conditions (S1 Fig). As expected for a native MDR1 substrate, over-expressing MDR1 increased cellular tolerance to Roc-A, and the effect was reversed by the third-generation MDR1 inhibitor elacridar (Fig 1D). We then performed the standard 44-h infection assay with HCT-8/MDR1, HCT-8/NC, and wild-type HCT-8 monolayers. The dose–response curves and $EC_{50}$ values for *C. parvum* were indistinguishable across the three host-cell backgrounds, indicating that elevated efflux in host cells does not diminish Roc-A's antiparasitic activity. These findings confirm that the compound's efficacy is attributable entirely to action on the parasite target rather than on a host target.

### In-vivo efficacy and oocyst-shedding assays

In-vivo activity was assessed in two immunodeficient mouse models of chronic *C. parvum* infection: interferon-γ-knockout (IFN-γ-KO) mice on a C57BL/6 background and NCG mice (NOD/ShiLtJGpt-Prkdc[em26Cd52]Il2rg[em26Cd22]/Gpt; GemPharmatech, Nanjing, China). Animals were housed in a double-layer isolator system (Hebei Linhai Metal Products, Baoding, China).

**Dose-finding and tolerance.** To establish a tolerated dose, twelve IFN-γ-KO mice (8 wk, mixed sex) received Roc-A at 0, 0.5, 1.0, or 2.0 mg/kg/d (split 0.25-, 0.5-, or 1.0-mg/kg gavages 12 h apart) for seven days. Roc-A was formulated in 0.5% DMSO in water. Body weight was recorded daily or in specified intervals, and mice were scored for fur condition, posture, and behavior (modified health-score scale, S3 Table) [51,52]. Because 0.5 mg/kg/d was well tolerated, this split dose was selected for efficacy studies.

**Infection and treatment.** Fifteen IFN-γ-KO mice and ten NCG mice (8 wk, mixed sex) were inoculated orally with $5 \times 10^4$ *C. parvum* oocysts and maintained for 10 days to establish chronic infection. On day 10 post-infection (dpi) animals were randomized to receive Roc-A (0.5 mg/kg/d), paromomycin (1,000 mg kg/d, IFN-γ-KO study only), or vehicle (0.5% DMSO–water); all treatments were given twice daily for five consecutive days (Fig 3A). Mice were weighed each morning before dosing and monitored clinically as above.

**Fecal collection.** For IFN-γ-KO mice, fecal pellets were collected daily from 3 to 15 dpi and once more 24 h after the final dose. For NCG mice, pellets were collected on 3, 5, 7 dpi; daily during treatment (10–15 dpi); and on 17, 21, and 30 dpi. Individual mice were placed in partitioned cardboard boxes for ~1 h to obtain fresh pellets.

**DNA extraction and qPCR quantification.** Pellets were weighed, suspended 1:1 in PBS, and stored overnight at 4°C. DNA was extracted from 200 µL homogenate using the TIANamp Stool DNA Kit (Tiangen, Beijing, China) with six freeze–thaw cycles to rupture oocyst walls; DNA was eluted in 50 µL. Oocyst burden was quantified by qPCR for Cp18S rDNA with the same primer pair used in the in-vitro assay. Standard curves were generated by spiking *Cryptosporidium*-negative feces with $10^2$–$10^6$ oocysts, processing identically, and plotting $C_T$ versus $\log_{10}$ oocyst number. Experimental values were interpolated from these curves.

All animal procedures were conducted in accordance with institutional guidelines, and study endpoints were euthanasia at 17 dpi (IFN-γ-KO) or 30 dpi (NCG).

## Molecular analysis of CpeIF4A and phylogenetic reconstruction

The *C. parvum* genome contains two DEAD-box helicases annotated in CryptoDB as eukaryotic initiation factor 4A: cgd1_880 ("Eukaryotic initiation factor 4A") and cgd7_3940 ("eIF4A-1 eukaryotic translation initiation factor 4A-1 RNA SFII helicase"). BLASTP searches against the NCBI protein database showed that the cgd1_880 product is most similar to human eIF4A-I/II, whereas cgd7_3940 aligns most closely with human eIF4A-III. InterProScan was used to confirm conserved motifs and ATP-/RNA-binding residues in both proteins.

To assign these helicases to the correct DEAD-box subfamilies, we performed a Bayesian-inference phylogeny that included the two *C. parvum* sequences, their orthologs from additional *Cryptosporidium* species, representative Apicomplexa, and human eIF4A-I/II and III. Protein sequences were aligned with MUSCLE v3.8.31; identical or incomplete entries and poorly aligned regions were removed, yielding 77 taxa and 335 unambiguously aligned positions. MrBayes v3.2.7 was run for 1,000,000 generations under a mixed amino-acid model with a proportion of invariant sites and a four-category gamma distribution; four independent Markov chains were sampled every 1,000 generations, and the first 25% of trees were discarded as burn-in before constructing the majority-rule consensus. The resulting topology grouped CpeIF4A with the eIF4A-I/II clade and CpeIF4A1 with the eIF4A-III clade, consistent with the BLAST and domain analyses.

For structural insight, a homology model of CpeIF4A bound to Roc-A, a polypurine RNA, and the ATP analog adenylyl-imidodiphosphate (AMPPNP) was generated with SWISS-MODEL, using the crystal structure of the human eIF4A1–AMPPNP–Roc-A–polypurine RNA complex (PDB 5ZC9) as template [36]. Structural comparisons and visualization were carried out in PyMOL and ChimeraX.

## Expression of recombinant CpeIF4A and biochemical assays

A codon-optimized fragment encoding CpeIF4A (cgd1_880) was synthesized (Sangon Biotech, Shanghai) and cloned into pMAL-c5X for production of an MBP fusion (MBP-CpeIF4A) in *Escherichia coli* BL21(DE3). Expression was induced

with 0.4 mM IPTG (isopropyl-beta-D-thiogalactopyranoside) at 16°C for ~16 h. Cells were lysed and the fusion protein was purified on amylose resin, then eluted with maltose according to the manufacturer's protocol (New England Biolabs). Purity was verified by 10% SDS-PAGE and concentration determined with a Bradford assay (BSA standard). An MBP tag expressed from the empty pMAL-c5X vector was purified in parallel for use as a negative control.

Helicase activity was measured with a fluorescence-resonance energy-transfer (FRET) assay (Fig 3G). The substrate was a 5′-overhanging duplex RNA comprising a 42-nt strand labeled at its 5′ end with 6-FAM (5′-CAUUAUCGGAUAGU GGAACCUAGCUUCGACUAUCGGAUAAUC-3′-6-FAM) annealed to a 24-nt strand bearing a 3′ Black Hole Quencher-1 (BHQ-1–5′-GAUUAUCCGAUAGUCG AAGCUAGG-3′) [53]. Unwinding separates the fluorophore from the quencher, producing a fluorescence increase.

Reactions (50 µL) contained 50 mM HEPES (pH 7.5), 100 mM NaCl, 2 mM $MgCl_2$, 2 mM DTT, 5 mM ATP (unless noted), 20 U RNasin, and 50 pmol duplex RNA. After pre-warming to 37°C, reactions were initiated by adding 30 pmol MBP-CpeIF4A; 30 pmol MBP alone served for background subtraction. Fluorescence (excitation 492 nm, emission 517 nm) was recorded every minute for 100 min on a StepOnePlus Real-Time PCR System (Applied Biosystems).

Inhibition kinetics were determined by repeating the assay with serial dilutions of Roc-A. Dose–response data were fit to a four-parameter sigmoidal model to derive $IC_{50}$ values:

$$Y = \frac{IC_{max} \times X^h}{IC_{50}^h + X^h}$$

where $Y$ is the response, $X$ is the drug concentration, $IC_{max}$ is the upper plateaus of the curve, $IC_{50}$ is the 50% effective concentration, and $h$ is the slope factor of the curve (Hill slope).

**Thermal-shift assay (TSA)**

Binding affinities between CpeIF4A and its ligands were measured by thermal-shift assay (TSA) as described previously [54,55]. Reactions (50 µL) were assembled in 96-well PCR plates with HEPES buffer (90 mM $Na_2HPO_4$, 10 mM $NaH_2PO_4$, 2 mM DTT, pH 7.8) containing 5 µM MBP-CpeIF4A and 20 µM SYPRO Orange, with or without ATP, duplex RNA, and/or Roc-A at the indicated concentrations. Plates were briefly vortexed, centrifuged at 10,000 g for 2 min, equilibrated for 2 min at room temperature, and transferred to a real-time PCR instrument. Fluorescence was recorded while the temperature increased from 40°C to 70°C at 1°C/min. Melting temperatures ($T_m$) were extracted from the fluorescence curves by Boltzmann fitting:

$$Y = Bottom + \frac{Top - Bottom}{1 + e^{\left(\frac{T_m - X}{Slope}\right)}}$$

Thermal shifts ($\Delta T_m$) were plotted against ligand concentration. Apparent dissociation constants ($K_d$) were obtained by fitting the $\Delta T_m$ against concentration data to a four-parameter sigmoidal model.

**Detection of native CpeIF4A by western blotting and immunofluorescence**

A polyclonal rabbit antibody was raised against the C-terminal peptide [20]GEIESNYDEIVEC[32], which is unique to CpeIF4A. A specific-pathogen-free rabbit received five immunizations: 100 µg peptide–KLH in Freund's complete adjuvant for the initial intradermal injection, followed by three intradermal boosts of 50 µg peptide in incomplete adjuvant, and a final intravenous boost of 20 µg peptide without adjuvant. Serum was collected and affinity-purified on nitrocellulose to which 100 µg peptide had been immobilized. Bound IgG was eluted with 200 mM glycine (pH 3.5, 150 mM NaCl, 0.05% Tween-20) and neutralized immediately with 1 M Tris base (final pH 7.0) [56,57].

For western blot analysis, excysted sporozoites were lysed in RIPA buffer containing protease inhibitors. Lysate corresponding to $7 \times 10^7$ sporozoites per lane was resolved by SDS-PAGE and transferred to nitrocellulose. Blots were blocked in 5% BSA/TBST for 1 h, probed with affinity-purified anti-CpeIF4A (1:5) for 1 h, washed, and incubated with horseradish peroxidase (HRP)-conjugated goat anti-rabbit IgG (Invitrogen). Signal was developed with enhanced chemiluminescence (Beyotime); all steps were at room temperature unless noted.

Immunofluorescence assays were performed on sporozoites excysted in the presence or absence of Roc-A. Parasites were fixed in 4% paraformaldehyde for 1 h, washed, and allowed to settle on poly-L-lysine–coated slides for 1 h. After permeabilization with 0.2% Triton X-100 (5 min) and blocking with 5% skim milk (1 h), slides were incubated with anti-CpeIF4A (1 h), washed, and labeled with Alexa Fluor 488–conjugated goat anti-rabbit IgG (1:2,000; 1 h). Nuclei were counterstained with 1 μg/mL DAPI (4′,6-diamidino-2-phenylindole). Slides were washed, mounted in antifade medium, sealed, and viewed on an Olympus DP72 microscope. TIFF images were processed in Adobe Photoshop, adjusting overall levels and gamma uniformly to enhance clarity without altering underlying data.

### Effect of Roc-A on translation and transcription during sporozoite excystation

**Protein synthesis.** To test whether blocking CpeIF4A curtails protein synthesis, we quantified total protein in excysting sporozoites. Oocysts ($10^6$ per well, 100 μL) were incubated for 2 h at 37°C in RPMI-1640 supplemented with 7.5 mg/mL taurocholic acid, in the absence or presence of Roc-A (0–10 μM). Preliminary qRT-PCR measurements of Cp18S rRNA showed that sporozoite viability was unaffected by Roc-A up to 100 μM (Fig 6A), and microscopy confirmed normal morphology and CpeIF4A localization at 10 μM (Fig 6B). After excystation, cultures were centrifuged to separate supernatants (secreted proteins) from pellets (non-secreted proteins). Pellets were lysed in 50 μL RIPA buffer containing protease inhibitors, incubated on ice for 10 min, vortexed, and cleared by centrifugation. Supernatants and lysates were mixed with SDS-PAGE sample buffer, resolved on 10% gels (20 μL/lane), and transferred to nitrocellulose. Blots were probed with a rabbit pan-*Cryptosporidium* antiserum, and band intensities were quantified in Fiji ImageJ2. Dose–response curves for total non-secreted proteins and for a prominent 28-kDa secreted protein were fitted by nonlinear regression to obtain $IC_{50}$ values.

**Transcript levels.** To determine whether Roc-A also alters transcription, free sporozoites ($2 \times 10^7$ per tube) obtained after 1 h excystation were resuspended in RPMI-1640 and exposed to 0.15 μM Roc-A ($\approx EC_{70}$) or 0.5% DMSO for 2 h at 25°C (n = 3 biological replicates each). Sporozoites were pelleted, resuspended in DEPC-treated water, snap-frozen in liquid nitrogen, and shipped on dry ice to Personalbio (Shanghai) for RNA-seq. Total RNA was extracted with TRIzol, and its quality was verified on a NanoDrop spectrophotometer. For each sample, 3 μg RNA was used to construct Illumina libraries: mRNA was captured on poly-T beads, fragmented, converted to first-strand cDNA with random primers, and second-strand cDNA was synthesized after RNase H treatment. After adapter ligation, 400–500 bp fragments were selected and PCR-enriched; library size and concentration were determined with an Agilent Bioanalyzer 2100 high-sensitivity DNA assay. Paired-end sequencing was performed on an Illumina NovaSeq 6000.

Raw reads were filtered for quality, then aligned to the *C. parvum* reference genome with HISAT2 v2.1.0. Gene counts were obtained with HTSeq v0.9.1 and converted to FPKM values. Differential expression was analyzed with DESeq v1.38.3; genes with $|\log_2(\text{fold change})| > 2$ and $p < 0.05$ (or $q < 0.05$ for adjusted values) were designated as differentially expressed.

### Data analysis and statistics

All biochemical and in vitro assays were performed in at least two independent experiments, each with three or more biological replicates; qRT-PCRs were run in duplicate. The mouse efficacy studies (IFN-γ-KO and NCG; n = 5 per treatment group) and the RNA-seq experiment were conducted once. Numerical data were analyzed in GraphPad Prism (version 10

or later) unless otherwise noted. Differential transcript expression was processed with the software packages listed above. Group comparisons used two-way analysis of variance (ANOVA) followed by the indicated multiple-comparison tests; specific tests are stated in the figure legends or text.

## Supporting information

**S1 Table. Transcript abundance comparison of *CpeIF4A* gene with *CpeIF4A1* and one of the DEAD-box (DDX) domain-containing proteins clustered with DDX19/25 proteins (cgd8_4750).**
(PDF)

**S2 Table. Proteomic abundance comparison of CpeIF4A with CpeIF4A1 and one of the DEAD-box (DDX) domain-containing proteins clustered with DDX19/25 proteins (cgd8_4750).**
(PDF)

**S3 Table. Mouse health scoring scales.**
(PDF)

**S4 Table. All original data used to build the graphs, including values to build figure panels 1B, 1C, 1D and 1E; 2A, 2B, 2C and 2D; 3B, 3C, 3D, 3E and 3F; 4D, 4E, 4H, 4I and 4J; 5A, 5B, 5C, 5D, 5E and 5F; 7A, 7E and 7F.**
(XLSX)

**S1 Fig. Transient over-expression of MDR1 in host cells as detected by immunofluorescence assay (IFA) using an anti-MDR1 antibody.** In this in vitro model, HCT-8 cells were transiently transfected with pCVM3 vector carrying MDR1 gene (pCMV3-MDR1; marked as HCT-8/MDR1). Negative control cells were transfected with blank pCMV3 vector (marked as HCT-8/NC). Both cell types were labeled with ant-MDR1 antibody at the same concentration, and images were taken with the same exposure time.
(PDF)

**S2 Fig. Evaluation of the drug tolerance in mice to Roc-A.** C57BL/6 mice (8-wk old; 3 mice/group) were administered by oral gavage with Roc-A at 0 (1% DMSO vehicle), 0.5, 1.0 and 2.0 mg/kg/d in a single daily dose for 7 days. **(A)** Per-cent daily weight gains in mice from 0 to 7 dpa (day post-administration). Bars indicate standard error of the mean (SEM). $* = p < 0.05$, $** = p < 0.01$, $*** = p < 0.001$, and $**** = p < 0.0001$ by Tukey's multiple comparison test (vs. vehicle control). **(B)** Mouse health scores calculated based on the survival, weight gains, fur condition, hunchbackedness, and animal attitude scales (also see Materials and Methods for detail).
(PDF)

**S3 Fig. Phylogenetic tree of CpeIF4A and closely related orthologs from various *Cryptosporidium* species, selected apicomplexan species, and humans.** The displayed is a consensus tree inferred by Bayesian inference (BI) analysis, which separates the sequences into seven clusters. Six clusters contain human orthologs, including eIF4A-I/II, eIF4A-III, DDX19/25, DDX39, DDX6, and DDX5/17. The seventh cluster appears unique to *Cryptosporidium* (absent in humans and other apicomplexans), but closely related to DDX5/17 subfamily proteins.
(PDF)

**S4 Fig. Immunostaining of CpeIF4A in intracellular *C. parvum* cultured with HCT-8 cells.** Host cell monolayers infected with *C. parvum* were fixed with paraformaldehyde (4%) and stained with affinity-purified anti-CpeIF4A antibody. Nuclei (Nuc) were counterstained with DAPI. The images show that CpeIF4A is distributed in the cytoplasm with stronger signals under the plasma membrane in varied developmental stages during merogony.
(PDF)

**S5 Fig. Structural model of CpeIF4A superimposed with human eIF4A-I in complex with the ATP analog AMPPNP, Roc-A and polypurine RNA.** CpeIF4A structure is predicted by Swiss-Model and superimposed with the crystal structure of the human eIF4A1 in complex with AMPPNP, polypurine RNA and Roc-A (PDB entry: 5ZC9). The superimposed structure shows high structural similarity, but subtle difference, between CpeIF4A and human eIF4A-I, predicting the binding of Roc-A at the "bi-molecular cavity" formed by eIF4A-I and a sharply bent pair of purines in the RNA. Also see Fig 6 for more detailed comparison at the Roc-A binding surface.
(PDF)

## Acknowledgments

We are grateful to Dr. Pingwei Li (Department of Biochemistry and Biophysics, Texas A&M University) for assistance with structural modeling, and to Ms. Chenchen Wang (Institute of Zoonosis, Jilin University) for technical support with immunostaining related experiments.

## Author contributions

**Conceptualization:** Meng Li, Jigang Yin, Guan Zhu.

**Data curation:** Meng Li, Dongqiang Wang, Guan Zhu.

**Formal analysis:** Meng Li, Dongqiang Wang, Guan Zhu.

**Funding acquisition:** Jigang Yin, Guan Zhu.

**Investigation:** Meng Li, Dongqiang Wang, Beibei Zou, Pingwei Li.

**Methodology:** Meng Li, Dongqiang Wang, Beibei Zou, Pingwei Li, Guan Zhu.

**Supervision:** Jigang Yin, Guan Zhu.

**Validation:** Dongqiang Wang, Guan Zhu.

**Visualization:** Meng Li, Pingwei Li, Guan Zhu.

**Writing – original draft:** Meng Li, Guan Zhu.

**Writing – review & editing:** Meng Li, Guan Zhu.

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
