## [Decision Letter · Decision Letter 0]

PPATHOGENS-D-25-00012

The protein translation initiation machinery is targetable for developing fast-killing therapeutics against the zoonotic Cryptosporidium parvum

PLOS Pathogens

Dear Dr. Zhu,

Thank you for submitting your manuscript to PLOS Pathogens. After careful consideration, we feel that it has merit but does not fully meet PLOS Pathogens's publication criteria as it currently stands. Therefore, we invite you to submit a revised version of the manuscript that addresses the points raised during the review process.

Please submit your revised manuscript within 60 days Mar 28 2025 11:59PM. If you will need more time than this to complete your revisions, please reply to this message or contact the journal office at plospathogens@plos.org. Please include the following items when submitting your revised manuscript:

We look forward to receiving your revised manuscript.

Kind regards,

Dominique Soldati-Favre

Section Editor

PLOS Pathogens

Dominique Soldati-Favre

Section Editor

PLOS Pathogens

 Sumita Bhaduri-McIntosh

Editor-in-Chief

PLOS Pathogens

orcid.org/0000-0003-2946-9497

 Michael Malim

Editor-in-Chief

PLOS Pathogens

orcid.org/0000-0002-7699-2064

**Additional Editor Comments (if provided):**

The reviewers identified several key weaknesses that need to be addressed to strengthen the manuscript. First, there is insufficient discussion of RocA's effects on mammalian cells, its broad anti-pathogen activity, and the potential for structural differences between eIF4A proteins in Cryptosporidium and humans that could guide specificity improvements. Additionally, the novelty and selectivity of RocA for CpeIF4A are overstated, requiring a deeper exploration of its broader activity and other potential targets in Cryptosporidium. Lastly, the in vivo mouse data lacks evaluation of recrudescence post-treatment, and the manuscript would benefit from leveraging established Cryptosporidium mouse models and/or resistance-conferring mutation studies to validate CpeIF4A as the target and assess the compound's long-term efficacy. Essentiality of CpeIF4A  for parasite does not need to  experimentally demonstrated however literature on this factor in other apicomplexan parasites should be more comprehensively covered.

**Journal Requirements:**

2) We notice that your supplementary Figures, and Tables are included in the manuscript file. Please remove them and upload them with the file type 'Supporting Information'. Please ensure that each Supporting Information file has a legend listed in the manuscript after the references list.

3) Please ensure that the funders and grant numbers match between the Financial Disclosure field and the Funding Information tab in your submission form. Note that the funders must be provided in the same order in both places as well. State what role the funders took in the study. If the funders had no role in your study, please state: "The funders had no role in study design, data collection and analysis, decision to publish, or preparation of the manuscript.".

**Reviewers' Comments:**

Reviewer's Responses to Questions

**Part I - Summary**

Reviewer #1: Meng Li et al. found that C. parvum eukaryotic initiation factor 4A (CpeIF4A) could be a target for the development of fast-killing anti-cryptosporidial therapeutics. In a series of experiments, they showed that Roc-A is a specific inhibitor of eIF4A and is highly effective against C. parvum growth in vitro and in vivo. They also found that Roc-A tightly binds to the CpeIF4A-RNA complex and suppresses protein synthesis in the parasite.

Overall, the manuscript provides a novel drug target, CpeIF4A, and a promising anti-cryptosporidial drug, Roc-A. It paves the way for the discovery of new anti-cryptosporidial drugs. There are still some suggestions for specific points shown below that may be useful to improve the manuscript.

Reviewer #2: The manuscript by Li et al describes the inhibition of Cryptosporidium parvum eukaryotic initiation factor 4A (CpeIF4A) by Rocaglamide-A (RocA) compound using biochemical, in vitro and in vivo assays. The topic of this study is important and timely since it is imperative to discover new drug targets for the development of much needed effective drugs against cryptosporidiosis. The authors provide strong biochemical data demonstrating the inhibition of CpeIF4A activity by RocA, disruption of protein synthesis in sporozoites, as well as the quick parasite killing potential of this compound in vitro at nanomolar concentration. Additionally, they tested this compound in IFN-g KO mice and found a reduction in oocyst production. Although these results are interesting, they are preliminary and several additional experiments are required to fully establish that Roc-A is a selective and specific inhibitor that is only targeting Cryptosporidium eIF4A. The in vivo data is not rigorous as no intestinal histological analysis has been provided, and the infection was monitored for only a limited time after the end of the drug-treatment period. Furthermore, the authors do not cite or discuss studies that have previously reported anti-parasitic activities of RocA in other parasites such as Leishmania and Plasmodium (PMID:36144626; PMID: 21417924). RocA has been reported to be an inhibitor of prohibitin and has been reported to target both asexual-blood stage and gametocyte stages in the malaria parasite (PMID: 37088154). The lack of mention/discussion about these relevant studies on anti-parasitic potential of RocA, and the possibility that this compound may be hitting other targets prohibitin or other RNA helicases) limits the novelty of this work. Additional experiments are therefore required to validate this drug target in Cryptosporidium.

Reviewer #3: Very interesting and exciting study reporting that a natural product, rocaglamide A (Roc-A) targeting the translation initiation factor eIF4A has potential as an anti-cryptosporidium therapeutic. The compound has low nM efficacy, very high selectivity indices, appears to specifically target the Cp eIF4A and is very effective against Cryptosporidium in IFN-gamma knock out mice. eIF4A is an RNA helicase, and the authors demonstrate that Roc-A inhibits the helicase activity of eIF4a, and binds with high affinity to eIF4A in the presence of RNA and ATP. Moreover they show that RocA inhibits protein synthesis in encysted sporozoites. Overall the study is well described, and the experiments are rigorous. My major concerns are that the mouse experiment ends only 24 hours after the last treatment so there is no evaluation of the possibility of recrudescence, and overall there needs to be more in depth discussion of Roc-A’s activities against human eIF4A and comparison to the results presented here, as well as a discussion of the feasibility of developing a lead compound with better selectivity for Cp over mammalian cells (based on the toxicity in mice)

**Part II – Major Issues: Key Experiments Required for Acceptance**

Reviewer #1: 1. In Figures 4B and C, you have shown the alignment of CpeIF4A1 and CpeIF4A2 with other orthologs of different species. In the manuscript and figure legends, you only mentioned CpeIF4A and Cp eIF4A1. I am struggling to find the information about the CpeIF4A2. Please make sure that these two figures are correctly labelled or described.

Reviewer #2: Further experiments are required to establish that Roc-A is a selective and specific inhibitor of CpeIF4A. Although this gene may be likely essential for parasite survival, it needs to be experimentally validated suing genetic approaches.

In the mammalian literature, it has been shows that Phe163Leu-Ile199Met mutations in eIF4A1 confer RocA-resistance in HEK293 cells (PMC3796129, PMC6386617). Are any of these mutations conserved in CpeIF4A that can be mutated for RocA resistance studies and genetic validation of this target? Is it possible that Roc-A has other targets in Cryptosporidium such as other DEAD-box RNA helices, prohibitin etc?

There are acute and chronic (NOD-SCID-gamma) mouse models of crytposporidiosis and reporter parasites available to test the efficacy of compounds and monitor any relapse after the compound administration period is over. These model systems and in vivo efficacy assays are now typically used in the Cryptosporidium research field. Therefore, testing the Roc-A inhibitor using any one of these models would provide strong evidence on the in vivo efficacy of this compound and guide future compound optimization and efficacy studies.

Reviewer #3: 1. Fig 3: The mouse experiment demonstrates the impressive rapid action of Roc A against Cp infection, but there is no evaluation of the possibility of recrudescence once treatment with RocA is stopped (final samples are taken 24 hours after last administration of drug). This is a concern as the RocA dose is extraordinarily low (0.5 mg/kg).

2. It would be helpful in the introduction/discussion to have more information on the effects of RocA on mammalian cells. A quick glace at PubMed reveals that there is a significant body of information on Roc-As effects on mammalian cells. While the in vitro selectivity indices are excellent, the toxicity in the mice is concerning, and the literature could provide insight into why this compound is so toxic in mice. Supplemental Fig 4A shows CpeIF4A predicted structure overlaid on the crystal structure of human eIF4A1-ATP-ROC-RNA complex, but there is little discussion of the potential to improve the specificity of the compound for Cp, beyond mentioning that the two IF4s have high structural similarity. The manuscript would be greatly enriched by a more in depth discussion of this compound’s activities against the human eIF4 and how this compares to the activity against CpeIF4A. For example, residues that confer resistance to RocA are found near the RNA binding motif (Sadlish, 2013). Are there resistance conferring residues in any of the apicomplexan sequences?

**Part III – Minor Issues: Editorial and Data Presentation Modifications**

Reviewer #1: 1. In figures 2C and 2D, why do you use 3 nM Roc-A to the culture, is it the value of EC90? Could you clarify the reason for using 3 nM Roc-A?

2. In Figure 2D, you treated Roc-A in culture for 3-10 h or 3-22 h, and both treatments could inhibit C. parvum growth on HCT-8 cells. But for these two treatment time points, the drug was effective on the asexual life stage of C. parvum. I was wondering if you have tried adding Roc-A at the sexual life stage (e.g., 30-44 h) to see if the Roc-A could still block parasite growth or not?

3. For the in vivo drug efficacy test (Figure 3C), is the data from one or two individual mouse experiments?

4. In Figure 4F, it would be better to add scale bars to the images.

5. Why did you choose sporozoites for the RNA-seq assay? Based on Figure 6B, I don't think there are significant differences in fluorescence signal when you treat sporozoites with/without Roc-A for 2 h. In addition, the transcription level in sporozoites is relatively lower than in other parasite stages. CpeIF4A is present at relatively high levels in all parasite life stages (Figure 4D). Have you tried RNA-seq in intracellular parasites? Maybe you should try to infect Cp on HCT-8 cells, treat with/without Roc-A for a few hours, and then do the RNA-seq. It might give you more candidates to analysis.

6. Has eukaryotic initiation factor 4A been studied in other apicomplexan parasites? It would be better to add some background about the eukaryotic initiation factor 4A from other apicomplexan parasites in the introduction or discussion section.

Reviewer #2: Line 127: HapG2 to be changed to ‘HepG2’

Line 523: MUSCL change to ‘MUSCLE’

Reviewer #3: Line 127: The approximate selectivity index should be 56,000 not 560,000

Fig 1D: and lines 139-140-While the text indicates that there is a significant dose dependent effect of elacridar on the cells, there is no statistical comparison between elacridar doses in Fig 1D.

Fig 1D and associated methods: What is the transfection efficiency of the human MDR1 gene in HCT8 cells?

Fig 4F: In addition to the IFA, this fig shows a western blot (not mentioned in the figure legend) that I presume shows the anti-CpeIF4A reactivity to the sporozoites lysate. There are 1 major and 5 minor bands recognized in the blot calling into question the specificity of the antibody. How specific is this antibody for CpeIF4A?

Fig 6C. I cannot find a description or a reference for the anti-Cryptosporidium antibody used in this experiment.

There are minor typo’s throughout that need to be fixed. Ie: line 127: HapG2 should be HepG2; line 327: the IC50s have no units, Supplemental fig 3: titles should read meronts not merozonts

PLOS authors have the option to publish the peer review history of their article (what does this mean? ). If published, this will include your full peer review and any attached files.

**Do you want your identity to be public for this peer review?** For information about this choice, including consent withdrawal, please see our Privacy Policy .

Reviewer #1: No

Reviewer #2: No

Reviewer #3: No

**Figure resubmission:**
---

## [Decision Letter · Decision Letter 1]

Dear Dr. Zhu,

We are pleased to inform you that your manuscript 'Targeting translation initiation yields fast-killing therapeutics against the zoonotic parasite Cryptosporidium parvum' has been provisionally accepted for publication in PLOS Pathogens.

This study identifies translation initiation via the parasite-specific eIF4A helicase as a previously unrecognized vulnerability in Cryptosporidium parvum, revealing Rocaglamide A as a potent and selective lead compound for fast-acting therapeutic intervention. all the reviewers' comments have been fully addressed. 

Best regards,

Dominique Soldati-Favre

Section Editor

PLOS Pathogens

Sumita Bhaduri-McIntosh

Editor-in-Chief

PLOS Pathogens

orcid.org/0000-0003-2946-9497

Michael Malim

Editor-in-Chief

PLOS Pathogens

orcid.org/0000-0002-7699-2064

Reviewer Comments (if any, and for reference):

Reviewer's Responses to Questions

**Part I - Summary**

Reviewer #1: The authors have addressed and clarified all concerns in the revised manuscript. I have no further comments on the study.

Reviewer #3: The paper is really excellent now with the additional experiemnts. All my concerns have been addressed

**Part II – Major Issues: Key Experiments Required for Acceptance**

Reviewer #1: I have no further questions on this study.

Reviewer #3: Additional experiments included in the revision have resolved any concerns of this reviewer

**Part III – Minor Issues: Editorial and Data Presentation Modifications**

Reviewer #1: I have no further questions on this study.

Reviewer #3: none

PLOS authors have the option to publish the peer review history of their article (what does this mean? ). If published, this will include your full peer review and any attached files.

**Do you want your identity to be public for this peer review?** For information about this choice, including consent withdrawal, please see our Privacy Policy .

Reviewer #1: No

Reviewer #3: No

---

## [Editor Report · Acceptance letter]

Dear Dr. Zhu,

We are delighted to inform you that your manuscript, "Targeting translation initiation yields fast-killing therapeutics against the zoonotic parasite Cryptosporidium parvum," has been formally accepted for publication in PLOS Pathogens.

Best regards,

Sumita Bhaduri-McIntosh

Editor-in-Chief

PLOS Pathogens

orcid.org/0000-0003-2946-9497

Michael Malim

Editor-in-Chief

PLOS Pathogens

orcid.org/0000-0002-7699-2064